# Fast Bayesian Inference for Gaussian Cox Processes via Path Integral Formulation

**Hideaki Kim**
NTT Human Informatics Laboratories
NTT Corporation
hideaki.kin.cn@hco.ntt.co.jp

## Abstract

Gaussian Cox processes are widely-used point process models that use a Gaussian process to describe the Bayesian a priori uncertainty present in latent intensity functions. In this paper, we propose a novel Bayesian inference scheme for Gaussian Cox processes by exploiting a conceptually-intuitive *path integral* formulation. The proposed scheme does not rely on domain discretization, scales linearly with the number of observed events, has a lower complexity than the state-of-the-art variational Bayesian schemes with respect to the number of inducing points, and is applicable to a wide range of Gaussian Cox processes with various types of link functions. Our scheme is especially beneficial under the multi-dimensional input setting, where the number of inducing points tends to be large. We evaluate our scheme on synthetic and real-world data, and show that it achieves comparable predictive accuracy while being tens of times faster than reference methods.

## 1 Introduction

Gaussian Cox processes constitute a class of doubly-stochastic point process models in which a flexible prior over a latent intensity function is established by a Gaussian process through a positive link function under which point events are generated. Gaussian Cox processes are the gold standard in analyzing event data in a Bayesian manner, and have a wide ranging list of applications in neuroscience [10], finance [5], and spatio-temporal analysis [30].

The observation process, i.e., the likelihood function of the Gaussian Cox process, depends on the functional form of the latent intensity function over a compact domain, which makes Bayesian inference challenging as it imposes the intractable computation of infinite-dimensional distributions. Many algorithms have been proposed to deal with this difficulty. They include Markov Chain Monte Carlo (MCMC) sampling, domain discretization, and variational Bayesian (VB) approximation. MCMC approaches [2, 22] provide exact inferencing for a specific link function, but they have excessive computation costs. Although domain discretization [10, 43] achieves sub-linear computational scaling with discretization size, it suffers from poor scaling with data size and the dimension of the domain. To overcome these limitations, state-of-the-art algorithms employ sparse VB approximation with inducing points [3, 31]; they achieve improved scaling with the dimension of the domain as well as computation cost that is linear to data size. However, the computation costs of VB approaches scale at the rate of $\mathcal{O}(NL^2 + L^3)$ where $L$ and $N$ are the number of inducing points and data points, respectively, making them problematic in the multi-dimensional domain setting, where $L$ tends to be large. Also, VB approaches achieve good approximations by aggressively focusing on the unique structures of individual kernel functions or link functions, and thus are not applicable to a wide range of Gaussian Cox processes.

In this paper, we introduce a conceptually-intuitive *path integral* formulation of Gaussian processes, by which we derive a novel inference scheme for tackling intractable Gaussian Cox processes. The

formulation predicts that the maximum *a posteriori* (MAP) estimation of a latent intensity function coincides with the problem of finding the most likely path that maximizes the functional posterior, whose solution can be obtained by solving an explicit integral equation. The approximate predictive covariance is also acquired by solving the corresponding integral equation. We propose an efficient algorithm to solve the integral equations; it offers $\mathcal{O}(NL + L^2)$ computation scaling, that is, lower complexity than the state-of-the-art variational Bayesian schemes with respect to the number of inducing points, as well as linear computation scaling with data size. Furthermore, our inference scheme via path integral holds for any kernel function and a wide range of link functions, and thus provides a practical methodology against Gaussian Cox processes that conventional approaches have yet to tackle successfully.

In Section 2, we introduce the path integral formulation of Gaussian processes, and construct a new Bayesian inference scheme for Gaussian Cox processes. In Section 3, we provide a practical methodology to perform inferencing. In Section 4, we outline related work. In Section 5, we perform comparative evaluations against reference methods on synthetic and real-world data, and confirm that our scheme achieves comparable predictive accuracy with substantial speed improvements over state-of-the-art algorithms. Finally, Section 6 states our conclusions.

## 2 Path Integral Formulation

### 2.1 Gaussian Cox Process

We assume that a latent function on a compact space, $\mathcal{T} \subset \mathbb{R}^D$, $x(\boldsymbol{t}) : \mathcal{T} \to \mathbb{R}$, is generated from a Gaussian Process (GP), denoted by $\mathcal{GP}(x(\boldsymbol{t})|\mu, k(\boldsymbol{t}, \boldsymbol{t}'))$, and observations are generated from a process under latent function $x(\boldsymbol{t})$, where $\mu$ and $k$ are the mean value and the kernel/covariance function for $x(\boldsymbol{t})$, respectively. Given data set $\mathcal{D}$ observed within $\mathcal{T}$, we consider the problem of obtaining the maximum *a posteriori* (MAP) estimator of $x(\boldsymbol{t})$ that maximizes the posterior probability

$$p(x(\boldsymbol{t})|\mathcal{D}) = \frac{p(\mathcal{D}|x(\boldsymbol{t})) \, \mathcal{GP}(x(\boldsymbol{t})|\mu, k)}{\int \mathscr{D}x(\boldsymbol{t}) \, p(\mathcal{D}|x(\boldsymbol{t})) \, \mathcal{GP}(x(\boldsymbol{t})|\mu, k)}, \tag{1}$$

where $p(\mathcal{D}|x(\boldsymbol{t}))$ is the likelihood function, and $\int \mathscr{D}x(\boldsymbol{t})$ in the denominator represents the integral over the function or the infinite-dimensional variable $x(\boldsymbol{t})$.

The GP prior, $\mathcal{GP}(x(\boldsymbol{t})|\mu, k)$, can be regarded as a normal distribution for an infinite-dimensional variable (see Section 2.2), and thus if the likelihood function depends only on a set of latent functions on finite $N$ training points, $p(\mathcal{D}|x(\boldsymbol{t})) = p(\mathcal{D}|\{x(\boldsymbol{t}_n)\}_{n=1}^N)$, then MAP estimation for an infinite-dimensional variable reduces to that for a finite $N$-dimensional variable by marginalizing out the variables on the points other than $\{x(\boldsymbol{t}_n)\}_{n=1}^N$; this inference is tractable and has been examined extensively (e.g. GP regression, GP classification). Here, we focus on a more challenging situation where the likelihood function includes a factor that depends on an integral of a latent function over $\mathcal{T}$ as follows,

$$\log p(\mathcal{D}|x(\boldsymbol{t})) = \sum_{n=1}^N \log \lambda(\boldsymbol{t}_n) - \int_{\mathcal{T}} \lambda(\boldsymbol{t}) d\boldsymbol{t}, \quad \lambda(\boldsymbol{t}) = \kappa\big(x(\boldsymbol{t})\big), \tag{2}$$

where $\mathcal{D} = \{\boldsymbol{t}_n\}_{n=1}^N$ is the set of point events occurring in the observation region $\mathcal{T}$, $\kappa(x) : \mathbb{R} \to \mathbb{R}_+$ is a non-negative function called the *link function*, and $\lambda(\boldsymbol{t})$ is the intensity function or the instantaneous probability of events occurring at each point in $\mathcal{T}$. The probabilistic model defined by (1-2) is called the *Gaussian Cox process*.

Bayesian inference (1) on situation (2) is often referred as "*doubly-intractable*", because it requires solving the integral of the stochastic function $x(\boldsymbol{t})$ as well as computing the intractable posterior probability of an infinite-dimensional variable. Below, we address the problem directly by introducing *path integral*, a mathematical tool familiar in the field of quantum physics, to treat probability distributions of stochastic functions properly and intuitively.

### 2.2 Gaussian Process Prior via Path Integral

In order to derive an explicit description of GP prior, we first review a well-known formulation of GP [7], and reconstruct its explicit form of probability density distribution by using a path integral representation.

Consider a model defined in terms of a linear combination of $M$ fixed basis functions, $\{\phi_m(\boldsymbol{t})\}_{m=1}^M$, such that

$$x(\boldsymbol{t}) = \mu + \sum_{m=1}^M \alpha_m \phi_m(\boldsymbol{t}) = \mu + \boldsymbol{\alpha}^\top \boldsymbol{\phi}(\boldsymbol{t}), \quad \boldsymbol{t} \in \mathcal{T}, \tag{3}$$

where $\boldsymbol{\alpha} \triangleq (\alpha_1, \cdots, \alpha_M)^\top$ is the $M$-dimensional weight vector, $\boldsymbol{\phi}(\boldsymbol{t}) \triangleq (\phi_1(\boldsymbol{t}), \cdots, \phi_M(\boldsymbol{t}))^\top$, and $\mu$ is a mean parameter. If $\boldsymbol{\alpha}$ is generated from an $M$-dimensional Gaussian with a diagonal covariance matrix of the form

$$p(\boldsymbol{\alpha}) = \mathcal{N}(\boldsymbol{\alpha}|\boldsymbol{0}, c^{-1}\boldsymbol{I}_M), \tag{4}$$

the joint probability distribution for any finite (even infinite) set of variables, $\boldsymbol{x} \triangleq (x(\boldsymbol{t}_1), \ldots, x(\boldsymbol{t}_J))^\top$ for $J \leq \infty$, follows the $J$-dimensional Gaussian given by

$$p(\boldsymbol{x})d\boldsymbol{x} = \mathcal{N}(\boldsymbol{x}|\mu\boldsymbol{1}, \boldsymbol{K})d\boldsymbol{x}, \quad \boldsymbol{K}_{jj'} = k(\boldsymbol{t}_j, \boldsymbol{t}_{j'}) \triangleq c^{-1}\boldsymbol{\phi}(\boldsymbol{t}_j)^\top \boldsymbol{\phi}(\boldsymbol{t}_{j'}), \tag{5}$$

where $k(\boldsymbol{t}, \boldsymbol{t}')$ is a semi-definite kernel function, $c$ is the precision parameter, $1 \leq j, j' \leq J$, and $\boldsymbol{1} \triangleq (1, \ldots, 1)^\top$. Because (5) holds for arbitrary numbers of dimensions, $J$, it provides an implicit representation of GP prior.

Next, based on the expression (5), we derive a more explicit representation of GP prior. For simplicity of explanation, we assume that the domain $\mathcal{T}$ is one-dimensional, $\boldsymbol{t} = t$ and $\mathcal{T} = [0, T]$, and mean $\mu$ is zero. Let $\mathcal{K}$ be the integral operator corresponding to $k(t, t')$, and $\mathcal{K}^*$ be the inverse operator of $\mathcal{K}$,

$$\mathcal{K}^{(*)}x(t) \triangleq \int_{\mathcal{T}} k^{(*)}(t, t')x(t')dt', \qquad \mathcal{K}^* k(t, s) = \int_{\mathcal{T}} k^*(t, t')k(t', s)dt' = \delta(t - s), \tag{6}$$

where $\delta(\cdot)$ represents the Dirac delta function. Note that $\mathcal{K}^*$ and $k^*(t, t')$ should be described by a differential operator because relation (6) indicates that $k(t, s)$ is the Green function [15] for the operator $\mathcal{K}^*$, but we do not go into detail here. Under the finite scenario (5) with $[t_0, t_J] = [0, T]$ and $J \gg 1$, the second equation in (6) can be approximated by $\boldsymbol{K}^*\boldsymbol{\Delta}\boldsymbol{K} = \boldsymbol{\Delta}^{-1}$, where $\boldsymbol{\Delta}_{jj'} = (t_j - t_{j-1})\delta_{jj'}$, $\boldsymbol{K}_{jj'}^* = k^*(t_j, t_{j'})$, and $\delta_{jj'}$ is the Kronecker delta. Then we can write down the $J$-dimensional Gaussian (5) by using $\boldsymbol{K}^*$ as

$$p(\boldsymbol{x})d\boldsymbol{x} = \sqrt{\frac{|\boldsymbol{\Delta}\boldsymbol{K}^*\boldsymbol{\Delta}|}{(2\pi)^J}} \exp\left[-\frac{1}{2}(\boldsymbol{\Delta}\boldsymbol{x})^\top \boldsymbol{K}^*(\boldsymbol{\Delta}\boldsymbol{x})\right]d\boldsymbol{x}. \tag{7}$$

By taking the limit of the division number $J \to \infty$ ($\boldsymbol{\Delta} \to \boldsymbol{0}$) for the finite-dimensional Gaussian measure (7), we achieve the following *path integral* representation of GP prior over the latent function $x(t)$:

$$\mathcal{GP}(x(t)|0, k)\mathscr{D}x \triangleq \lim_{J \to \infty} p(\boldsymbol{x})\prod_{j=1}^J dx(t_j) = \sqrt{\frac{1}{|\mathcal{K}|}} \exp\left[-\frac{1}{2}\iint_{\mathcal{T}\times\mathcal{T}} k^*(t, s)x(t)x(s)dtds\right]\mathscr{D}x, \tag{8}$$

where $|\mathcal{K}| \triangleq \lim_{J \to \infty} |\boldsymbol{K}\boldsymbol{\Delta}|$, $\mathscr{D}x \triangleq \lim_{J \to \infty} \prod_{j=1}^J \sqrt{(t_j - t_{j-1})/(2\pi)}dx(t_j)$, and the following path integral holds,

$$\int \exp\left[-\frac{1}{2}\iint_{\mathcal{T}\times\mathcal{T}} k^*(t, s)x(t)x(s)dtds\right]\mathscr{D}x = \sqrt{|\mathcal{K}|}. \tag{9}$$

$|\mathcal{K}|$ represents the Fredholm determinant or functional determinant [19] of the integral operator $\mathcal{K}$, which is defined by the product of its eigenvalues. See the supplementary material (§1) for the derivation. The expression for $\mathcal{GP}(x(\boldsymbol{t})|\mu, k)$ is easily recovered by $x(t) \to x(t) - \mu$ and $t \to \boldsymbol{t}$ in (8).

The path integral representation (8-9) has a clear advantage over the standard one (5) in that the distribution of the latent function $x(\boldsymbol{t})$ is written in terms of the explicit integral of $x(\boldsymbol{t})$ over the domain $\mathcal{T}$, as with the likelihood function (2), which makes it possible to apply functional analyses, *calculus of variation*, to the Bayesian estimation (1). To the best of our knowledge, this is the first proposal of a path integral representation for GP prior in the machine learning community.

It should be noted that our derivation of the path integral representation is based on an intuitive view reminiscent of Feynman's path integral formulation of the quantum field theory [17], and thus is not mathematically rigorous. In this paper, we confirm that our path integral expression is valid by using experiments rather than taking a more mathematically rigorous approach; we discuss this briefly in Section 4.

## 2.3 Maximum *A Posteriori* Estimation

Using the representation (8) and the likelihood function (2), we can rewrite the posterior (1) as the functional,

$$p(x(\boldsymbol{t})|\mathcal{D})\mathscr{D}x = \frac{1}{p(\mathcal{D})}\exp\Big[-S\big(x(\boldsymbol{t}),\underline{x}(\boldsymbol{t})\big)\Big]\mathscr{D}x, \qquad (10)$$

where $S\big(x(\boldsymbol{t}),\underline{x}(\boldsymbol{t})\big)$ is an *action integral*, defined by

$$S\big(x(\boldsymbol{t}),\underline{x}(\boldsymbol{t})\big) \triangleq \int_{\mathcal{T}}\Big[\frac{1}{2}(x(\boldsymbol{t})-\mu)\underline{x}(\boldsymbol{t}) + \kappa\big(x(\boldsymbol{t})\big) - \sum_{n=1}^{N}\log\kappa\big(x(\boldsymbol{t})\big)\delta(\boldsymbol{t}-\boldsymbol{t}_n)\Big]d\boldsymbol{t} + \frac{1}{2}\log|\mathcal{K}|, \quad (11)$$

and $\underline{x}(\boldsymbol{t}) \triangleq \int_{\mathcal{T}}k^*(\boldsymbol{t},\boldsymbol{t}')(x(\boldsymbol{t}')-\mu)d\boldsymbol{t}' = \mathcal{K}^*(x-\mu)$. (10) indicates that MAP estimation of $x(\boldsymbol{t})$ is equivalent to the problem of finding the most likely function or path $x(\boldsymbol{t})$ that minimizes the action integral. Thus we now apply calculus of variations to the action integral, where the functional derivative of $S(x(\boldsymbol{t}),\underline{x}(\boldsymbol{t}))$ on the MAP estimator $\hat{x}(\boldsymbol{t})$ is equal to zero: $\frac{\delta S}{\delta\hat{x}(\boldsymbol{t})}\delta\hat{x}(\boldsymbol{t}) + \frac{\delta S}{\delta\hat{\underline{x}}(\boldsymbol{t})}\delta\hat{\underline{x}}(\boldsymbol{t}) = 0$. This leads us to realize the following equation for deriving the MAP estimator $\hat{x}(\boldsymbol{t})$,

$$\hat{x}(\boldsymbol{t}) + \int_{\mathcal{T}}k(\boldsymbol{t},\boldsymbol{t}')\dot{\kappa}\big(\hat{x}(\boldsymbol{t}')\big)d\boldsymbol{t}' = \mu + \sum_{n=1}^{N}k(\boldsymbol{t},\boldsymbol{t}_n)\frac{\dot{\kappa}\big(\hat{x}(\boldsymbol{t}_n)\big)}{\kappa\big(\hat{x}(\boldsymbol{t}_n)\big)}, \quad \boldsymbol{t}\in\mathcal{T}, \qquad (12)$$

where $\dot{\kappa}(x) \triangleq d\kappa(x)/dx$. See the supplementary material (§2) for the detailed derivations of (12).

## 2.4 Predictive Covariance

One of the advantages of GP models over non-Bayesian approaches is that they can provide predictive distributions. We apply a Laplace approximation to GP models (10), and find the approximate form of the predictive covariance.

We now know the mode of the posterior, $\hat{x}(\boldsymbol{t})$, and consider a Taylor expansion of functional action potential $S\big(x(\boldsymbol{t}),\underline{x}(\boldsymbol{t})\big)$ centered on the mode such that

$$S\big(x(\boldsymbol{t}),\underline{x}(\boldsymbol{t})\big) \simeq S\big(\hat{x}(\boldsymbol{t}),\hat{\underline{x}}(\boldsymbol{t})\big) + \frac{1}{2}\iint_{\mathcal{T}\times\mathcal{T}}\sigma^*(\boldsymbol{t},\boldsymbol{s})(x(\boldsymbol{t})-\hat{x}(\boldsymbol{t}))(x(\boldsymbol{s})-\hat{x}(\boldsymbol{s}))d\boldsymbol{t}d\boldsymbol{s}, \qquad (13)$$

where $\sigma^*(\boldsymbol{t},\boldsymbol{s}) \triangleq \frac{\delta^2 S(x,\underline{x})}{\delta x(\boldsymbol{t})\delta x(\boldsymbol{s})}\big|_{x=\hat{x}}$ is the second derivative of $S$. The first term in the Taylor expansion vanishes due to the stationary condition. The quadratic approximation of the action integral corresponds to the approximation of the posterior process by a GP, and the predictive covariance or the kernel function for the posterior GP, denoted by $\sigma(\boldsymbol{t},\boldsymbol{s})$, can be obtained by the functional inversion of $\sigma^*(\boldsymbol{t},\boldsymbol{s})$ (see Equation (8)), which results in

$$\sigma(\boldsymbol{t},\boldsymbol{s}) = h(\boldsymbol{t},\boldsymbol{s}) - \boldsymbol{h}(\boldsymbol{t})^\top(\boldsymbol{Z}+\boldsymbol{H})^{-1}\boldsymbol{h}(\boldsymbol{s}), \qquad (14)$$

where $\boldsymbol{Z}_{nn'} \triangleq \delta_{nn'}\frac{\kappa^2(x)}{\dot{\kappa}^2(x)-\kappa(x)\ddot{\kappa}(x)}\big|_{x=\hat{x}(\boldsymbol{t}_n)}$, $\ddot{\kappa}(x) \triangleq \frac{d^2\kappa(x)}{dx^2}$, $\boldsymbol{H}_{nn'} \triangleq h(\boldsymbol{t}_n,\boldsymbol{t}_{n'})$, $\boldsymbol{h}(\boldsymbol{t}) \triangleq (h(\boldsymbol{t},\boldsymbol{t}_1),\ldots,h(\boldsymbol{t},\boldsymbol{t}_N))^\top$, and $h(\boldsymbol{t},\boldsymbol{s})$ is defined by the following Fredholm integral equation of the second kind [36],

$$h(\boldsymbol{t},\boldsymbol{s}) + \int_{\mathcal{T}}k(\boldsymbol{t},\boldsymbol{t}')\ddot{\kappa}\big(\hat{x}(\boldsymbol{t}')\big)h(\boldsymbol{t}',\boldsymbol{s})d\boldsymbol{t}' = k(\boldsymbol{t},\boldsymbol{s}). \qquad (15)$$

Note that $\boldsymbol{Z}_{nn'}$ has an infinite value for $k(x)=e^x$, which leads to the relation, $\sigma(\boldsymbol{t},\boldsymbol{s})=h(\boldsymbol{t},\boldsymbol{s})$. The full derivation of (14-15) is given in the supplementary material (§3).

## 2.5 Marginal Likelihood

Let $\Sigma$ and $\mathcal{H}$ be the integral operators for $\sigma(\boldsymbol{t},\boldsymbol{s})$ and $h(\boldsymbol{t},\boldsymbol{s})$, respectively: $\Sigma \triangleq \int_{\mathcal{T}}\cdot\sigma(\boldsymbol{t},\boldsymbol{t}')d\boldsymbol{t}'$, $\mathcal{H} \triangleq \int_{\mathcal{T}}\cdot h(\boldsymbol{t},\boldsymbol{t}')d\boldsymbol{t}'$. Under Laplace approximation (13), we can obtain the marginal likelihood or evidence of Gaussian Cox processes, $p(\mathcal{D})$, by performing the following path integral (9),

$$\log p(\mathcal{D}) = \log\int\exp\Big[-S\big(x(\boldsymbol{t}),\underline{x}(\boldsymbol{t})\big)\Big]\mathscr{D}x \simeq -S\big(\hat{x}(\boldsymbol{t}),\hat{\underline{x}}(\boldsymbol{t})\big) + \frac{1}{2}\log|\Sigma|, \qquad (16)$$

Table 1: Link functions $\kappa(x)$ and their derivation.

| | $\kappa(x)$ | $\dot{\kappa}(x)$ | $\ddot{\kappa}(x)$ | $\gamma(\dot{\kappa}) = \dot{\kappa}/\kappa$ |
|---|---|---|---|---|
| quadratic | $x^2$ | $2x$ | $2$ | $4/\dot{\kappa}$ |
| exponential | $\exp(x)$ | $\exp(x)$ | $\exp(x)$ | $1$ |
| softplus | $\log(1+\exp(x))$ | $(1+\exp(-x))^{-1}$ | $\exp(-x)/(1-\exp(-x))^2$ | $-\log(1-\dot{\kappa})$ |

where $\log|\Sigma| = \log|\mathcal{H}| - \log|\boldsymbol{I}_N + \boldsymbol{Z}^{-1}\boldsymbol{H}|$, and

$$S\big(\hat{x}(\boldsymbol{t}),\hat{\underline{x}}(\boldsymbol{t})\big) = \frac{1}{2}\log|\mathcal{K}| + \sum_{n=1}^{N}\left[\frac{1}{2}(\hat{x}(\boldsymbol{t}_n)-\mu)\frac{\dot{\kappa}\big(\hat{x}(\boldsymbol{t}_n)\big)}{\kappa\big(\hat{x}(\boldsymbol{t}_n)\big)} - \log\kappa\big(\hat{x}(\boldsymbol{t}_n)\big)\right]$$

$$+ \int_{\mathcal{T}}\kappa\big(\hat{x}(\boldsymbol{t})\big)d\boldsymbol{t} - \frac{1}{2}\int_{\mathcal{T}}(\hat{x}(\boldsymbol{t})-\mu)\dot{\kappa}\big(\hat{x}(\boldsymbol{t})\big)d\boldsymbol{t}. \tag{17}$$

The full derivation is given in the supplementary material (§4).

## 3 How to Solve Integral Equations

To obtain the MAP estimator $\hat{x}(\boldsymbol{t})$ and the predictive covariance $\sigma(\boldsymbol{t},\boldsymbol{s})$, we need to solve the corresponding integral equations. We provide efficient algorithms to solve them (also see Suppl. (§8)).

### 3.1 MAP Estimator

Equation (12) for the MAP estimator is a nonlinear integral equation of the second kind, sometimes called the Hammerstein integral equation [23]. We can obtain an approximation solution of the equation through the most widely used class of methods, *projection methods* [4], which approximate the solution or its derivation by choosing an approximation from a given finite $L$-dimensional linear subspace of functions, denoted by $\mathcal{Z}$. We employ the *collocation method* [29], a variant of the projection method:

$$\dot{\kappa}\big(\hat{x}(\boldsymbol{t})\big) \simeq \sum_{l=1}^{L}\hat{\beta}_l\varphi_l(\boldsymbol{t}), \quad \{\hat{\beta}_l\}_{l=1}^{L} = \operatorname*{arg\,min}_{\{\beta_l\}_{l=1}^{L}}\sum_{l=1}^{L}[r(\boldsymbol{p}_l)]^2,$$

$$r(\boldsymbol{t}) \triangleq \dot{\kappa}\bigg(\sum_n k(\boldsymbol{t},\boldsymbol{t}_n)\gamma\Big[\sum_l \beta_l\varphi_l(\boldsymbol{t}_n)\Big] - \sum_l \beta_l\int_{\mathcal{T}}k(\boldsymbol{t},\boldsymbol{t}')\varphi_l(\boldsymbol{t}')d\boldsymbol{t}' + \mu\bigg) - \sum_l \beta_l\varphi_l(\boldsymbol{t}), \tag{18}$$

where $r(\boldsymbol{t})$ is the residual derived from (12), $\{\varphi_l\}_{l=1}^{L}$ is a basis for $\mathcal{Z}$, $\{\boldsymbol{p}_l\in\mathcal{T}\}_{l=1}^{L}$ are the collocation points, and $\gamma(\dot{\kappa})$ represents $\dot{\kappa}(x)/\kappa(x)$ being re-written as a function of $\dot{\kappa}(x)$ (see Table 1). As the basis $\{\varphi_l\}_{l=1}^{L}$, we adopt the eigenfunctions of the kernel operator $\mathcal{K}$, which reduces the problematic integral term in the residual into a tractable one as $\int_{\mathcal{T}}k(\boldsymbol{t},\boldsymbol{t}')\varphi_l(\boldsymbol{t}')d\boldsymbol{t}' = \lambda_l\varphi_l(\boldsymbol{t})$, where $\{\lambda_l\}_{l=1}^{L}$ is the set of the eigenvalues of $\mathcal{K}$. The procedure for finding the eigenfunctions is provided at the end of this section. For fair comparison with VB-based approaches of Gaussian Cox processes [3, 31], we solve the minimization problem of the sum of the squared residuals by using a popular gradient descent algorithm, *Adam* [27]. We can obtain the MAP estimator $\hat{x}(\boldsymbol{t})$ by substituting (18) into (12): $\hat{x}(\boldsymbol{t}) = \mu + \sum_n k(\boldsymbol{t},\boldsymbol{t}_n)\gamma(\sum_l\hat{\beta}_l\varphi_l(\boldsymbol{t}_n)) - \sum_l\lambda_l\hat{\beta}_l\varphi_l(\boldsymbol{t})$.

The sparsely located collocation points, $\{\boldsymbol{p}_l\}_{l=1}^{L}$, reduce the computational complexity of MAP estimation substantially, and play a similar role in the Bayesian inference scheme to that played by the *inducing points* in the variational framework for sparse GP regression [44]. In this paper, we adopt a regular grid over $\mathcal{T}$ as the collocation points in accordance with the VB-based references [31, 3], but note that other location choices are possible. For simplicity of explanation, we hereafter call $\{\boldsymbol{p}_l\}_{l=1}^{L}$ inducing points.

It should be noted here that our approach (18) with the collocation method can be applied to cases where $\gamma(\dot{\kappa}) = \dot{\kappa}(x)/\kappa(x)$ is defined properly as a function of $\dot{\kappa}(x)$. Table 1 shows that $\gamma(\dot{\kappa})$ can be defined for popular link functions which include quadratic [18, 31], exponential [12, 30, 33], and softplus [28] functions. But our approach is not applicable to link functions whose derivatives, $\dot{\kappa}(x)$, are not monotonic with respect to $x$, such as sigmoidal link functions [2, 3, 22].

## 3.2 Predictive Covariance

Given the MAP estimator $\hat{x}(\boldsymbol{t})$, the predictive covariance (14) can be obtained by solving the linear integral equation (15) about $h(\boldsymbol{t}, \boldsymbol{s})$. We apply the *Gelerkin method* [4], a variant of the projection method, to solve the equation, which results in

$$h(\boldsymbol{t}, \boldsymbol{s}) \simeq \sum_l \omega_l \varphi_l(\boldsymbol{t}) \varphi_l(\boldsymbol{s}), \quad \omega_l = \frac{\lambda_l}{1 + \lambda_l \Xi_l}, \quad \Xi_l = \int_{\mathcal{T}} \ddot{\kappa}(\hat{x}(\boldsymbol{t})) \varphi_l^2(\boldsymbol{t}) d\boldsymbol{t}. \tag{19}$$

See the supplementary material (§5) for the detailed derivation. We estimate the integral in $\{\Xi_l\}_{l=1}^L$ via the Monte Carlo method, but the quadratic link function ($\kappa(x) = x^2$) is an exceptional case in which the integration can be performed analytically, $\Xi_l = 2$, due to the orthogonal relation of the eigenfunctions, $\int_{\mathcal{T}} \varphi_l(\boldsymbol{t}) \varphi_{l'}(\boldsymbol{t}) d\boldsymbol{t} = \delta_{ll'}$. It should be noted that (19) is the Mercer expansion [32] of function $h(\boldsymbol{t}, \boldsymbol{s})$, where $\{\omega_l\}_{l=1}^L$ is a set of its eigenvalues.

## 3.3 Marginal Likelihood

We can evaluate the marginal likelihood by substituting (18) into (16-17),

$$\log p(\mathcal{D}) = \sum_n \log \kappa(\hat{x}(\boldsymbol{t}_n)) - \int_{\mathcal{T}} \kappa(\hat{x}(\boldsymbol{t})) d\boldsymbol{t} - \frac{1}{2} \sum_l \lambda_l \beta_l^2$$
$$+ \frac{1}{2} \sum_n \frac{\dot{\kappa}(\hat{x}(\boldsymbol{t}_n))}{\kappa(\hat{x}(\boldsymbol{t}_n))} \left[ \sum_l \lambda_l \beta_l \varphi_l(\boldsymbol{t}_n) - (\hat{x}(\boldsymbol{t}_n) - \mu) \right] - \frac{1}{2} \log |\boldsymbol{I}_N + \boldsymbol{Z}^{-1} \boldsymbol{H}| - \frac{1}{2} \log \frac{|\mathcal{K}|}{|\mathcal{H}|}, \tag{20}$$

where the second term, $\int_{\mathcal{T}} \kappa(\hat{x}(\boldsymbol{t})) d\boldsymbol{t}$, is estimated by the Monte Carlo method. The last term is associated with the functional determinant (see the supplementary material (§1)), which can be calculated by the product of its eigenvalues as, $\log \frac{|\mathcal{K}|}{|\mathcal{H}|} = \sum_{l=1}^L \log \frac{\lambda_l}{\omega_l} = \sum_{l=1}^L \log(1 + \lambda_l \Xi_l)$, where we approximate the kernel function in terms of the Mercer expansion up to the top $L$ eigenvalues, $k(\boldsymbol{t}, \boldsymbol{s}) \simeq \sum_{l=1}^L \lambda_l \varphi_l(\boldsymbol{t}) \varphi_l(\boldsymbol{s})$.

## 3.4 Computational Complexity

The objective function to be minimized in MAP estimation is $\sum_{l=1}^L [r(\boldsymbol{p}_l)]^2$, which needs the computation of $\mathcal{O}(NL + L^2)$ for each evaluation (see Eq. (18)): the computational complexities of $\{\gamma_n \triangleq \gamma [\sum_l \beta_l \varphi_l(\boldsymbol{t}_n)]\}_{n=1}^N$ and $\{\sum_n k(\boldsymbol{p}_l, \boldsymbol{t}_n) \gamma_n\}_{l=1}^L$ are both $\mathcal{O}(NL)$, and $\{\sum_l \lambda_l \beta_l \varphi_l(\boldsymbol{p}_{l'})\}_{l'=1}^L$ or $\{\sum_l \beta_l \varphi_l(\boldsymbol{p}_{l'})\}_{l'=1}^L$ needs the computation of $\mathcal{O}(L^2)$. When a gradient descent algorithm with automatic differentiation [6] is employed, the computational complexity of the MAP estimation per gradient descent step is equal to $\mathcal{O}(NL + L^2)$, which is one-$L$th the complexity of the VB-based approaches [31, 3]. We adopt the algorithm in Section 5.

Furthermore, in view of the shape of the objective function, we can substantially improve the computational efficiency of the MAP estimation by using a mini-batch gradient descent (MGD) algorithm: MGD reduces the computational complexity to $\mathcal{O}(NL)$, which is due to $\{\gamma_n\}_{n=1}^N$. We verify the effectiveness of MGD in the supplementary material (§10). Also, when $\{\boldsymbol{p}_l\}_l$ are located regularly over $\mathcal{T}$ and $k(\boldsymbol{t}, \boldsymbol{t}')$ is separable across the dimensions, matrix $\boldsymbol{\Phi}$ defined as $\boldsymbol{\Phi}_{ll'} = \varphi_l(\boldsymbol{p}_{l'})$ has the Kronecker structure (see the the supplementary material (§7)), which can further reduce the complexity to $\mathcal{O}(NL)$ under (batch) gradient descent. Note that the cost of the VB-based methods [31, 3] become $\mathcal{O}(NL^2)$ when the Kronecker structure of the gram matrix is exploited.

Once the MAP estimator or the set of coefficient $\{\beta_l\}_{l=1}^L$ is obtained, estimating the predictive covariance (14) needs the computation of $\mathcal{O}((N + L) \cdot \min(N^2, L^2) + N_{\mathrm{mc}} L)$, where $N_{\mathrm{mc}}$ is the number of samples for the Monte Carlo method. The cost $\mathcal{O}(N_{\mathrm{mc}} L)$ comes from the computation of $h(\boldsymbol{t}, \boldsymbol{s})$ (19), and the rest is due to the matrix operation. Due to the fact that $h(\boldsymbol{t}, \boldsymbol{s})$ is a degenerate kernel of rank $L$, the $N \times N$ matrix $\boldsymbol{H}$ can be decomposed into a product of $N \times L$ matrix $\boldsymbol{R}$, defined by $\boldsymbol{R}_{nl} = \sqrt{\omega_l} \varphi_l(\boldsymbol{t}_n)$, and its transpose as $\boldsymbol{H} = \boldsymbol{R} \boldsymbol{R}^{\top}$; The matrix operation $\boldsymbol{h}(\boldsymbol{t})^{\top} (\boldsymbol{Z} + \boldsymbol{R} \boldsymbol{R}^{\top})^{-1} \boldsymbol{h}(\boldsymbol{s})$ costs $\mathcal{O}(LN^2 + N^3)$ in a direct manner, but $\mathcal{O}(L^2 N + L^3)$ if the Woodbury matrix identity is used. Because the predictive covariance computation is not part of the iterative optimization in MAP estimation, its computation cost is negligible compared to that of MAP estimation. Also, the selection of hyper-parameters can be performed based on the marginal likelihood (20), which needs the computation of $\mathcal{O}((N + L) \cdot \min(N^2, L^2) + N_{\mathrm{mc}})$.

## 3.5 Eigenfunctions of Kernel Operator

The pairs of eigenvalues and eigenfunctions, $\{\lambda_l, \varphi_l\}_{l=1}^{\infty}$, for the kernel operator $\mathcal{K}$ are obtained by solving a homogeneous Fredholm integral equation, $\int_{\mathcal{T}} k(\boldsymbol{t}, \boldsymbol{s}) \varphi_l(\boldsymbol{s}) d\boldsymbol{s} = \lambda_l \varphi_l(\boldsymbol{t})$, where the set of eigenvalues is at most countable [36]. Here we assume a multiplicative kernel, $k(\boldsymbol{t}, \boldsymbol{s}) = \prod_{d=1}^{D} k^{(d)}(t^{(d)}, s^{(d)})$, and a hyper-rectangular domain with interval $\mathcal{T}^{(d)} = [0, T^{(d)}]$ in each dimension $d$. The multidimensional integral equation can then be reduced to a set of unidimensional ones, $\{\int_{\mathcal{T}^{(d)}} k^{(d)}(t^{(d)}, s) \varphi_{l_d}^{(d)}(s) ds = \lambda_{l_d}^{(d)} \varphi_{l_d}^{(d)}(t^{(d)})\}_d$, via the separation of variables, which results in a multiplicative form of solution, $(\lambda_{\boldsymbol{l}}, \varphi_{\boldsymbol{l}}(\boldsymbol{t})) = \left( \prod_d \lambda_{l_d}^{(d)}, \prod_d \varphi_{l_d}^{(d)}(t^{(d)}) \right)$, where $(\lambda_{l_d}^{(d)}, \varphi_{l_d}^{(d)})$ is the $l_d$-th solution of the integral equation in dimension $d$, and $\boldsymbol{l} = (l_1, \ldots, l_D)$. Thus the computation complexity of solving the integral equation scales linearly with the number of dimensions, $D$. We adopt the eigenfunctions with the top $L_d$ eigenvalues in each dimension $d$ as the basis for the projection method, resulting in a set of $L = \prod_d L_d$ basis functions.

The integral equation for each dimension $d$ can rarely be solved analytically, but we can approximate the solution by the Nyström method [35]: The integral term is approximated by a $J$-point numerical integration such that $\sum_{j=1}^{J} k(t, s_j) \tilde{\varphi}_l(s_j) w = \tilde{\lambda}_l \tilde{\varphi}_l(t)$, where $(\tilde{\lambda}_l, \tilde{\varphi}_l)$ is the approximation solution, $\{s_j\}_{j=1}^{J}$ and $w \triangleq \int_{\mathcal{T}} 1 dt / J$ denote the nodes and the weight, respectively, and index $d$ is omitted for simplicity of explanation (e.g. $k^{(d)}(t^{(d)}, s^{(d)}) \rightarrow k(t, s)$); the approximation solution is then obtained by using the eigenvalues $e_l$ and eigenvectors $\boldsymbol{v}_l$ of the $J \times J$ matrix $\tilde{K}$, defined by $\tilde{K}_{jj'} \triangleq k(s_j, s_{j'})$, as

$$\tilde{\lambda}_l = e_l w, \quad \tilde{\varphi}_l(t) = \tilde{\boldsymbol{k}}(t)^{\top} \boldsymbol{v}_l / (e_l \sqrt{w}), \tag{21}$$

where $\tilde{\boldsymbol{k}}(t) \triangleq (k(t, s_1), \ldots, k(t, s_J))^{\top}$. Under the Nyström method, the computational complexity of finding the eigenfunctions is $\mathcal{O}(DJ^3)$. We set $J = 1000$ in this paper, which is negligible compared to the cost of the MAP estimation.

# 4 Related Work

**Gaussian Cox Processes:** Various algorithms have been proposed to deal with the "*doubly-intractable*" difficulty of inferencing in Gaussian Cox processes (GCPs). The classical approach is domain discretization [10, 12, 33, 38, 43]; it approximates the integral in the likelihood function using a Riemann sum by assuming that the intensity function is constant over each grid. Although domain discretization is applicable to general GCPs regardless of link/kernel function, it suffers from poor scaling in terms of the data size and domain dimension, as well as being sensitive to discretization size. The MCMC approaches [2, 22] provide exact inferencing by exploiting the unique structure of the sigmoidal link function, but they are not applicable to large-scale datasets because it demands $\mathcal{O}(N^3)$ computations. State-of-the-art algorithms employ the sparse VB approximation with inducing points [3, 31], and achieve computation cost linear to the data size, $\mathcal{O}(NL^2 + L^3)$, where $L$ is the number of inducing points. However, the VB approaches remain problematic in a multi-dimensional domain setting, where the number of inducing points tends to be large. Also, they exploit the unique structures of individual link functions (i.e. quadratic [31] and sigmoidal [3]) to obtain feasible algorithms, which limits their application. An approach closely related to ours is the sparse MAP-Laplace method developed by Donner and Opper [13]; it restricts the link function to being sigmoidal and demands Monte Carlo integration to perform MAP estimation. By focusing on the unique structure of quadratic link function and Matèrn kernel function, John and Hensman proposed a MCMC-based method [24] that has the same complexity as our model. The merit of [24] is that it allows for a non-Gaussian posterior over latent function to improve the approximation quality, while our method is beneficial in that it's deterministic and applicable to various link and kernel functions. Our focus is on a deterministic approach with the most standard Gaussian kernel function, but it is an important next step to make a comparative analysis with the scalable MCMC-based method. A summary of the related works about the complexity is provided in the supplementary material (§9).

Under the quadratic link function, our MAP estimation (12) involves a representer theorem, and the exact MAP estimator with the equivalent kernel [42] can be derived (see the supplementary material (§6)). The same MAP estimator was derived by Flaxman et al. for the regularized maximum likelihood problem [18], while Walder and Bishop derived its predictive covariance through Mercer's

theorem [45]. Our scheme, with its path integral formulation, is a generalization of their works in that our scheme holds (i) for general link functions and (ii) for non-zero mean parameter $\mu \neq 0$.

**Path Integral Formulation:** The path integral formulation of GCPs (10) is almost the same as that of quantum mechanics (QM) as posited by Richard Feynman [17], except for the imaginary factor in QM's action potential. In QM, the MAP estimator equation equals the Euler-Lagrange equation, and the Laplace approximation (13) for functional distribution corresponds to a kind of semi-classical calculation of QM solutions. The connection between GCPs (or generally GP models) and QM suggests that physics-inspired techniques developed in QM could be applied to Bayesian inferencing in machine learning. In statistical physics, for example, Chang et al. have proposed to apply a path integral approach to the Bayesian inference problem in membrane biophysics [9].

Our GP formulation (8) agrees with the probability function expressed by the Onsager-Machlup functional (OMF) [8, 16] for linear stochastic differential equation (SDE), which indicates that the inverse kernel function $k^*(t, t')$ could be described by using the linear differential operator, $\mathcal{L} = \sum_n c_n (d/dt)^n$, of the corresponding SDE. This is consistent with the fact that the kernels in GPs are strongly related to differential operators [37]. However, our approach differs from OMF in terms of how they represent the MAP estimator: our estimator (12) is described by means of kernel function $k(t, t')$, while the OMF approach obtains the estimator by maximizing the posterior described by $\mathcal{L}$. The advantage of the former is, as in the case of "kernel trick" in the linear regression problem, that it does not need to know the explicit form of $\mathcal{L}$, which would otherwise require an infinite sum of differentials for the closed form of $k(t, t')$. The benefit of the latter is that it is applicable to nonlinear SDEs, which is out of scope for GP.

A possible approach to make our derivation of path integral formulation (8) clearer in measure-theoretic terms is introducing the abstract Wiener measure [21], which is closely related to the reproducing kernel Hilbert space [25] and can be used to define Gaussian measure in an infinite-dimensional space. Alternatively, the Radon-Nikodym derivative is a promising tool for justifying the path integral expression with the aid of the fertile stream of works by Andrew Stuart and co-authors [11].

## 5 Experiments

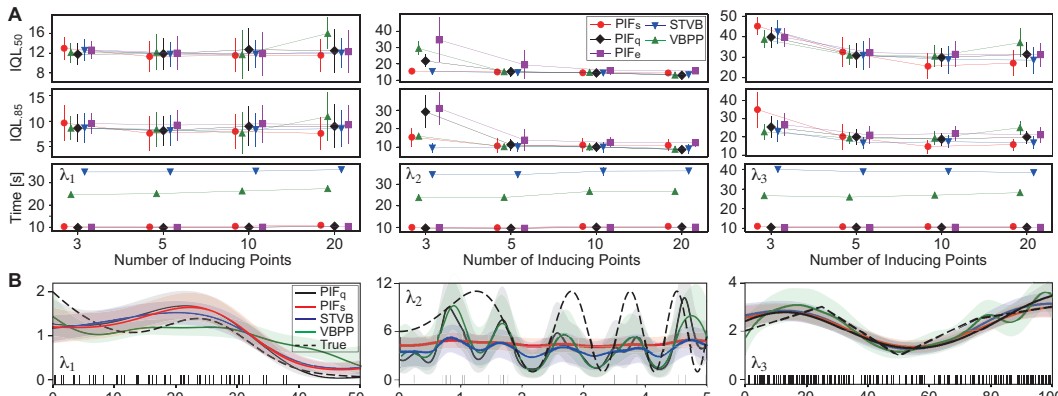

Figure 1: Results on three types of synthetic data. (A) The predictive performances and the CPU times. Lower values are better. The error bars represent the standard deviations. (B) The estimated intensity functions ($L = 20$). Solid lines and shaded areas represent MAP estimators and [0.15, 0.85] prediction intervals, respectively.

We examined the validity and the potential efficiency of our path integral formulation (PIF) by evaluating it against deterministic and scalable VB-based inference approaches on synthetic and open real-world data. We adopted the state-of-the-art structured variational inference of sigmoidal GCP [3] (STVB), and the variational inference of GCP with quadratic link function [31] (VBPP) as the references. For our proposal, we employed GCPs with exponential (PIF$_e$), quadratic (PIF$_q$), and softplus (PIF$_s$) link functions. This time we used a multiplicative Gaussian kernel, $k(t, s) = \prod_d e^{-(\theta(t_d - s_d))^2}$, where the hyper-parameter $\theta$ was optimized by maximizing the

marginal likelihood (16) or the evidence lower bound [3, 31] through grid search. For fair comparison, we performed estimations using *Adam* [27] with 5,000 iterations across all methods. Each of the CPU times reported is the amount of time required to calculate the MAP, the predictive covariance, and the marginal likelihood given a hyper-parameter, where computing the eigenfunctions is included. A MacBook Pro with 4-core CPU (2.8 GHz) was used. For details, see the supplementary material (§10).

## 5.1 Synthetic Data

In accordance with [2], we created 1D data sets generated from three types of intensity functions: $\lambda_1(t) = 2\exp(-t/15) + \exp(-[(t - 25)/10]^2)$ for $t \in [0, 50]$; $\lambda_2(t) = 5\sin(t^2) + 6$ for $t \in [0, 5]$; $\lambda_3(t)$ is a piecewise linear function in Figure 1 for $t \in [0, 100]$, each of which has 20 trial sequences. For each trial, the intensity function was estimated by each of the methods, and the performance was evaluated based on the integrated $\rho$-*quantile loss* [41], defined as $\mathrm{IQL}_\rho \triangleq \int_{\mathcal{T}} 2(\lambda(t) - \hat{\lambda}(t))(\rho \mathrm{I}_{\lambda(t) > \hat{\lambda}(t)} - (1 - \rho)\mathrm{I}_{\lambda(t) \leq \hat{\lambda}(t)}) dt$, where I, $\hat{\lambda}(t)$, and $\lambda(t)$ denote the indicator, the predicted $\rho$-quantile of the intensity function, and the true one, respectively. Here, we adopted $\mathrm{IQL}_{.5}$ (integrated absolute error) and $\mathrm{IQL}_{.85}$.

Figure 1A displays the predictive performances as functions of the number of inducing points $L$; it shows that our approaches (PIF) matched the performances of the VB-based methods across the three data sets. In particular, the comparison between $\mathrm{PIF}_q$ and VBPP is informative because both used the same kernel and link functions but relied on different approximations. VB-based approximations are usually better at estimating posterior distributions than Laplace approximations, but Figure 1A shows that our $\mathrm{PIF}_q$ with Laplace approximation was comparable to VBPP under $L \gtrsim 5$ with regard to both $\mathrm{IQL}_{.85}$ and $\mathrm{IQL}_{.50}$, which demonstrates the practical utility for recovering the posterior distribution as well as point estimation.

Figure 1A also plots the CPU times needed for estimation, showing that PIF can be performed several times faster than the scalable VB-based methods. Figure 1A-B shows that when $L$ is large, the quadratic link function ($\mathrm{PIF}_q$, VBPP) is preferable for the highly-modulated scenario of $\lambda_2(t)$, while the sigmoidal and softplus ones (STVB, $\mathrm{PIF}_s$) work better on $\lambda_1(t)$ and $\lambda_3(t)$. Link functions should be carefully selected depending on the data, and thus it is an advantage of our PIF that it supports scalable estimation algorithms for various link functions within the same scheme.

## 5.2 Real-world Data

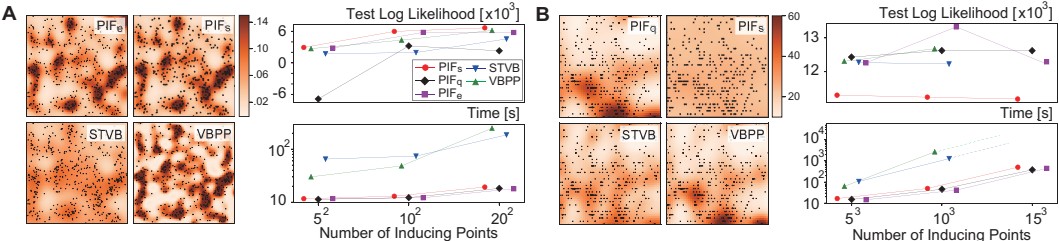

Figure 2: The estimated intensity functions, the predictive performances, and the CPU times on open real-world data. Higher values of log likelihoods and lower values of CPU times are better. (A) 2D neuronal data. (B) 3D spatio-temporal taxi data. The CPU times of STVB and VBPP exceeded 10 hours with $L = 15^3$, and the estimations were given up.

We examined the validity of our approach based on multi-dimensional real-world data sets. One is a 2D neuronal data, where event locations correspond to the position of a mouse moving in an arena with recorded cell firing [39, 40] (CC-BY); We adopted the training ($N_{\mathrm{train}} = 583$) and the test ($N_{\mathrm{test}} = 29127$) data as randomly assigned by [3]. The other is a spatio-temporal 3D taxi dataset in the city of Porto [34], where the pick-up locations and times are considered as observations (no personally identifiable information is included, CC-BY); We adopted the training ($N_{\mathrm{train}} = 1000$) and the test ($N_{\mathrm{test}} = 3401$) data used by [3]. We ran the models on the training data, and evaluated their predictive performances based on the test log likelihood. Note that to eliminate the effect of data assignment

bias, the evaluation multiplied estimated intensity by a factor, $N_{\text{test}}/N_{\text{train}}$. We used a regular grid over the multi-dimensional domain $\mathcal{T}$ as the inducing points.

Figure 2 displays the predictive and time performances as functions of $L$, and shows that our PIF-based approaches achieved comparable predictive accuracy while being tens of times faster than the VB-based methods. All the methods typically required a relatively large number of inducing points, $L$, to achieve good performance, which suggests that our approaches are especially beneficial in the multi-dimensional domain settings because they have lower complexity with $L$ than the VB-based alternatives. Stochastic optimization algorithms could further reduce the complexity, which is demonstrated in the supplementary material (§10). Note here that the memory complexity of our proposed method, $\mathcal{O}(NL + L^2)$, is the same as that of the VB-based approaches. In the 3D taxi dataset, PIF$_s$ output the intensity function with the smallest volatility among the compared methods (see Figure 2B), which led to the poor performance. This implies that the softplus link function might tend to underestimate the intensity function modulation given sparsely located data points, which highlights the importance of appropriately selecting the link function. Also, to clarify that our method is scalable to data size, we ran additional experiments on a larger ($N \simeq 10^5$) taxi dataset, see the supplementary material (§10).

# 6 Discussions

**Conclusions:** We have proposed a novel *path integral* formulation of Gaussian Cox processes (GCPs), by which we have derived a scalable inference scheme that holds for a wide range of GCPs with various types of link functions. Based on synthetic and open real-world data, we confirmed that our scheme achieves comparable predictive accuracy while being substantially faster than conventional alternatives.

**Future work & limitations:** We focused on investigating the practicality of our scheme, and our derivation of the path integral representation is intuitive but not mathematically rigorous. Making our derivation of path integral formulation clearer, especially in measure-theoretic terms, is an important task for future work. Also, we adopted the collocation method to solve the MAP (integral) equation, but exploiting other algorithms for solving integral equations, such as the Nyström method [35] and the degenerate kernel method [26], might yield further improvements in computational efficiency.

We did not consider optimizing hyper-parameters with gradient descent algorithms in this paper, but VB-based approaches allow for optimization with gradient methods, which are particularly beneficial when the kernel function has several kinds of hyper-parameters. It is worth noting that our approach can also employ gradient methods: because the objective function, $L(\theta, \hat{\boldsymbol{\beta}}(\theta)) \equiv \log p(\mathcal{D}|\theta)$, has an argmin, $\hat{\boldsymbol{\beta}}(\theta) \equiv \{\hat{\beta}_l(\theta)\}_{l=1}^{L}$ (see Eqs. (18-20)), the hyper-parameter optimization in our approach belongs to the bi-level optimization, and the exact computation of the gradient of $L(\theta, \hat{\boldsymbol{\beta}}(\theta))$ can be executed [20]; Here we show only the result, $\frac{dL(\theta, \hat{\boldsymbol{\beta}}(\theta))}{d\theta} = \frac{\partial L}{\partial \theta} - \left(\nabla_{\boldsymbol{\beta}} L\right)^{\top} \left(\nabla_{\boldsymbol{\beta}\boldsymbol{\beta}}^2 f\right)^{-1} \left(\frac{\partial}{\partial \theta} \nabla_{\boldsymbol{\beta}} f\right)$, where $f(\theta, \hat{\boldsymbol{\beta}}) \equiv \sum_l [r(\boldsymbol{p}_l)]^2$. When automatic differentiation is employed, the complexity of computing the gradient is equal to the sum of the complexity of the marginal likelihood and $\mathcal{O}(L^3)$, where $\mathcal{O}(L^3)$ comes from matrix inversion. Here, we note that the gradient of the eigenfunctions and eigenvalues obtained by the Nyström method with respect to the hyper-parameter can be evaluated analytically. When a gradient descent algorithm is employed for hyper-parameter optimization, our approach needs to alternate between the optimization in the $L$-dimensional space (MAP estimation) and that in the dim($\theta$)-dimensional space, while VB-based approaches perform the optimization in higher ($L^2 + \dim(\theta)$)-dimensional space. Although it is out-of-scope of this paper, a comparative analysis between our approach and VB-based alternatives to investigate the practical utility is an important next step in future work.

**Broader impact:** Our proposed scheme via the path integral representation can be applied to not only GCPs, but also a wide range of Gaussian process models whose likelihood functions include problematic integrals of latent functions. Gaussian process density models (GPDMs) [1, 14], for example, come under such models, and our scheme would provide a new tangible inference algorithm for GPDMs different from MCMC or VB-based algorithms. Although our scheme itself does not contain any ethical problems nor negative societal impacts, applying GCPs for predicting fine spatio-temporal patterns of people's behaviors might harm their privacy in some cases, and thus great care should be taken to protect personal information.

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
