# Supplementary Material
## "Fast Bayesian Inference for Gaussian Cox Processes via Path Integral Formulation"

**Hideaki Kim**
NTT Human Informatics Laboratories
NTT Corporation
hideaki.kin.cn@hco.ntt.co.jp

## S1 Functional Determinant

Here we show that the functional determinant of integral operator $\mathcal{K}$, defined by $|\mathcal{K}| \triangleq \lim_{J \to \infty} |\boldsymbol{K}\boldsymbol{\Delta}|$ in (8), can be calculated by the product of its eigenvalues.

Under the discrete scenario (7), the $l$-th eigenvalue of the $J \times J$ matrix, $\boldsymbol{K}\boldsymbol{\Delta}$, should satisfy the following eigenvalue equation,

$$\sum_{j'=1}^{J} k(t_j, t_{j'})(t_{j'} - t_{j'-1})\tilde{v}_{j'}^l = \tilde{\lambda}_l \tilde{v}_j^l, \quad \forall j \in \{1, \ldots, J\}, \tag{S1}$$

where $\tilde{\lambda}_l$ denotes the $l$-th eigenvalue, and $\tilde{\boldsymbol{v}}^l \triangleq (\tilde{v}_1^l, \ldots, \tilde{v}_J^l)^\top$ is the $J$-element eigenvector for the $l$-th eigenvalue. The determinant of $\boldsymbol{K}\boldsymbol{\Delta}$ is thus given by the product of the finite number of eigenvalues, $|\boldsymbol{K}\boldsymbol{\Delta}| = \prod_{l=1}^{L} \tilde{\lambda}_l$. Given the limit of the division number $J \to \infty$ ($\Delta \to \boldsymbol{0}$), the Nyström method [8] states that the eigenvalue equation (S1) converges to a homogeneous Fredholm integral equation, i.e., the eigenvalue equation of operator $\mathcal{K}$,

$$\mathcal{K}v^l = \lambda_l v^l, \quad \text{or} \quad \int_{\mathcal{T}} k(t,s)v^l(s)ds = \lambda_l v^l(t), \; t \in \mathcal{T}, \tag{S2}$$

where $v^l(t_j) = \tilde{v}_j^l$, and $\lambda_l = \lim_{J \to \infty} \tilde{\lambda}_l$. The set of eigenvalues for the integral equation (S2) is countably infinite [9]. Therefore, the functional determinant of operator $\mathcal{K}$, that is, $|\mathcal{K}| \triangleq \lim_{J \to \infty} |\boldsymbol{K}\boldsymbol{\Delta}|$, can be described as the product of the infinite number of eigenvalues of (S2):

$$|\mathcal{K}| \triangleq \lim_{J \to \infty} |\boldsymbol{K}\boldsymbol{\Delta}| = \prod_{l=1}^{\infty} \lambda_l. \tag{S3}$$

It should be noted that integral operator $\mathcal{K}$ with a positive definite kernel function often has infinitely small positive eigenvalues, where the marginalization of Gaussian process prior by path integral (9) leads to the trivial value of zero, $\sqrt{|\mathcal{K}|} = 0$. Actually, the path integral provides a rational result when applied to the marginalization of the posterior probability (16), where the ratio of two functional determinants are considered.

35th Conference on Neural Information Processing Systems (NeurIPS 2021).

## S2 Derivation of MAP Estimator Equation

We detail here the derivation of the integral equation (12) that yields the MAP estimator. The functional derivative of $S(x(t), \underline{x}(t))$ should be zero on the MAP estimator $\hat{x}(t)$:

$$
\delta S(\hat{x}(t), \underline{\hat{x}}(t)) = \int_{\mathcal{T}} \left[ \frac{\delta S}{\delta \hat{x}(t)} \delta x(t) + \frac{\delta S}{\delta \underline{\hat{x}}(t)} \delta \underline{x}(t) \right] dt + O((\delta x)^2)
$$

$$
\simeq \int_{\mathcal{T}} \left[ \dot{\kappa}(\hat{x}(t)) - \sum_{n=1}^{N} \frac{\dot{\kappa}(\hat{x}(t_n))}{\kappa(\hat{x}(t_n))} \delta(t - t_n) + \frac{1}{2} \underline{\hat{x}}(t) \right] \delta x dt + \int_{\mathcal{T}} \frac{1}{2} (\hat{x}(t) - \mu) \delta \underline{x} dt
$$

$$
= \int_{\mathcal{T}} \left[ \dot{\kappa}(\hat{x}(t)) - \sum_{n=1}^{N} \frac{\dot{\kappa}(\hat{x}(t_n))}{\kappa(\hat{x}(t_n))} \delta(t - t_n) + \underline{\hat{x}}(t) \right] \delta x dt = 0,
$$

where the following relation was used,

$$
\int_{\mathcal{T}} (\hat{x}(t) - \mu) \delta \underline{x} dt = \int_{\mathcal{T}} (\hat{x}(t) - \mu) \int_{\mathcal{T}} k^*(t, t') \delta x(t') dt' dt
$$

$$
= \int_{\mathcal{T}} dt' \delta x(t') \int_{\mathcal{T}} k^*(t, t')(\hat{x}(t) - \mu) dt
$$

$$
= \int_{\mathcal{T}} \underline{\hat{x}}(t') \delta x dt'. \quad \because) \ k^*(t, t') = k^*(t', t)
$$

Thus the following equation is derived,

$$
\underline{\hat{x}}(t) + \dot{\kappa}(\hat{x}(t)) = \sum_{n=1}^{N} \frac{\dot{\kappa}(\hat{x}(t_n))}{\kappa(\hat{x}(t_n))} \delta(t - t_n), \quad t \in \mathcal{T}. \tag{S4}
$$

By applying operator $\mathcal{K}$ to (S4), we realize an equation that derives the MAP estimator $\hat{x}(t)$ as follows,

$$
\hat{x}(t) + \int_{\mathcal{T}} k(t, t') \dot{\kappa}(\hat{x}(t')) dt' = \mu + \sum_{n=1}^{N} k(t, t_n) \frac{\dot{\kappa}(\hat{x}(t_n))}{\kappa(\hat{x}(t_n))}, \quad t \in \mathcal{T}. \tag{S5}
$$

## S3 Derivation of Predictive Covariance

We detail the derivation of the predictive covariance shown in (13-15). The predictive inverse co-variance (precision), denoted by $\sigma^*(t, t')$, is given by the second functional derivative of $S$, which is written as

$$
\sigma^*(t, t') \triangleq \left. \frac{\delta^2 S(x, \underline{x})}{\delta x(t) \delta x(t')} \right|_{x=\hat{x}} = z^*(t, t') + h^*(t, t'),
$$

where

$$
z^*(t, t') \triangleq -\sum_{n=1}^{N} \left. \frac{d^2 \log(\kappa(x))}{dx^2} \right|_{x=\hat{x}(t)} \delta(t - t_n) \delta(t' - t_n),
$$

$$
h^*(t, t') \triangleq a(t, t') + k^*(t, t'), \tag{S6}
$$

$$
a(t, t') \triangleq \ddot{\kappa}(\hat{x}(t)) \delta(t - t').
$$

Let the integral operators corresponding to $\sigma(t, t')$, $z(t, t')$, $h(t, t')$, and $a(t, t')$ be denoted by $\Sigma$, $\mathcal{Z}$, $\mathcal{H}$, and $\mathcal{A}$, respectively, and their inverse counterparts by $^*$. Using the fact that operator $\mathcal{Z}^*$ is factorized as,

$$
\mathcal{Z}^* = \int_{\mathcal{T}} \cdot \, z^*(t, t') dt' = \mathcal{U}^\top \mathbf{Z}^{-1} \mathcal{U},
$$

$$
\mathbf{Z}_{nn'} \triangleq -\left. \frac{d^2 \log(\kappa(x))}{dx^2} \right|_{x=\hat{x}(t)} \delta_{nn'}, \quad \mathcal{U}_n \triangleq \int_{\mathcal{T}} \cdot \, \delta(t' - t_n) dt',
$$

we obtain the predictive covariance $\sigma(\boldsymbol{t}, \boldsymbol{t}')$ with a finite (thus tractable) $N$-dimensional matrix representation as follows:

$$
\begin{aligned}
\Sigma = \int_{\mathcal{T}} \cdot\, \sigma(\boldsymbol{t}, \boldsymbol{t}')d\boldsymbol{t}' &= (\mathcal{Z}^* + \mathcal{H}^*)^* = (\mathcal{U}^\top \boldsymbol{Z}^{-1}\mathcal{U} + \mathcal{H}^*)^* \\
&= \mathcal{H} - \mathcal{H}\mathcal{U}^\top (\boldsymbol{Z} + \mathcal{U}\mathcal{H}\mathcal{U}^\top)^* \mathcal{U}\mathcal{H} \\
&= \int_{\mathcal{T}} \cdot\, \Big[ h(\boldsymbol{t}, \boldsymbol{t}') - \boldsymbol{h}(\boldsymbol{t})^\top (\boldsymbol{Z} + \boldsymbol{H})^{-1}\boldsymbol{h}(\boldsymbol{t}') \Big] d\boldsymbol{t}',
\end{aligned}
$$

$$
\therefore\ \ \sigma(\boldsymbol{t}, \boldsymbol{t}') = h(\boldsymbol{t}, \boldsymbol{t}') - \boldsymbol{h}(\boldsymbol{t})^\top (\boldsymbol{Z} + \boldsymbol{H})^{-1}\boldsymbol{h}(\boldsymbol{t}'),
$$

where $\boldsymbol{H}_{nn'} \triangleq h(\boldsymbol{t}_n, \boldsymbol{t}_{n'})$, $\boldsymbol{h}(\boldsymbol{t}) \triangleq (h(\boldsymbol{t}, \boldsymbol{t}_1), \dots, h(\boldsymbol{t}, \boldsymbol{t}_N))^\top$; we used the Woodbury matrix identity in this derivation.

The remaining problem is how to obtain $h(\boldsymbol{t}, \boldsymbol{t}')$. Equation (S6) states that the operators $\mathcal{H}$, $\mathcal{A}$, and $\mathcal{K}$ hold the relation, $\mathcal{H}^* = \mathcal{A} + \mathcal{K}^* \Leftrightarrow (\mathcal{I} + \mathcal{K}\mathcal{A})\mathcal{H} = \mathcal{K}$, which leads to the integral equation that $h(\boldsymbol{t}, \boldsymbol{t}')$ should satisfy,

$$
h(\boldsymbol{t}, \boldsymbol{s}) + \int_{\mathcal{T}} k(\boldsymbol{t}, \boldsymbol{t}')\ddot{\kappa}\big(\hat{x}(\boldsymbol{t}')\big)h(\boldsymbol{t}', \boldsymbol{s})d\boldsymbol{t}' = k(\boldsymbol{t}, \boldsymbol{s}). \tag{S7}
$$

## S4   Derivation of Marginal Likelihood

We can obtain the marginal likelihood or evidence of GP models, $p(\mathcal{D})$, under Laplace approximation (13) by performing path integral (9) as follows:

$$
\begin{aligned}
\log p(\mathcal{D}) &= \log \int \exp\big[-S(x(\boldsymbol{t}), \underline{x}(\boldsymbol{t}))\big]\mathscr{D}x \\
&\simeq -S(\hat{x}(\boldsymbol{t}), \hat{\underline{x}}(\boldsymbol{t})) + \log \int e^{-\frac{1}{2}\iint_{\mathcal{T}\times\mathcal{T}} \sigma^*(\boldsymbol{t}, \boldsymbol{t}')(x(\boldsymbol{t})-\hat{x}(\boldsymbol{t}))(x(\boldsymbol{t}')-\hat{x}(\boldsymbol{t}'))d\boldsymbol{t}d\boldsymbol{t}'}\mathscr{D}x \\
&= -S(\hat{x}(\boldsymbol{t}), \hat{\underline{x}}(\boldsymbol{t})) + \frac{1}{2}\log|\Sigma|.
\end{aligned}
$$

Substituting (S4) into (11), we can write down $S(\hat{x}(\boldsymbol{t}), \hat{\underline{x}}(\boldsymbol{t}))$ as

$$
\begin{aligned}
S(\hat{x}(\boldsymbol{t}), \hat{\underline{x}}(\boldsymbol{t})) &= \frac{1}{2}\log|\mathcal{K}| - \sum_{n=1}^{N}\log\kappa\big(\hat{x}(\boldsymbol{t}_n)\big) + \int_{\mathcal{T}}\kappa\big(\hat{x}(\boldsymbol{t})\big)d\boldsymbol{t} + \frac{1}{2}\int_{\mathcal{T}}(\hat{x}(\boldsymbol{t}) - \mu)\hat{\underline{x}}(\boldsymbol{t})d\boldsymbol{t} \\
&= \frac{1}{2}\log|\mathcal{K}| - \sum_{n=1}^{N}\log\kappa\big(\hat{x}(\boldsymbol{t}_n)\big) + \int_{\mathcal{T}}\kappa\big(\hat{x}(\boldsymbol{t})\big)d\boldsymbol{t} \\
&\qquad + \frac{1}{2}\int_{\mathcal{T}}(\hat{x}(\boldsymbol{t}) - \mu)\bigg[\sum_{n=1}^{N}\frac{\dot{\kappa}\big(\hat{x}(\boldsymbol{t}_n)\big)}{\kappa\big(\hat{x}(\boldsymbol{t}_n)\big)}\delta(\boldsymbol{t} - \boldsymbol{t}_n) - \dot{\kappa}\big(\hat{x}(\boldsymbol{t})\big)\bigg]d\boldsymbol{t} \\
&= \frac{1}{2}\log|\mathcal{K}| + \int_{\mathcal{T}}\kappa\big(\hat{x}(\boldsymbol{t})\big)d\boldsymbol{t} - \frac{1}{2}\int_{\mathcal{T}}(\hat{x}(\boldsymbol{t}) - \mu)\dot{\kappa}\big(\hat{x}(\boldsymbol{t})\big)d\boldsymbol{t} \\
&\qquad + \sum_{n=1}^{N}\bigg[\frac{1}{2}(\hat{x}(\boldsymbol{t}_n) - \mu)\frac{\dot{\kappa}\big(\hat{x}(\boldsymbol{t}_n)\big)}{\kappa\big(\hat{x}(\boldsymbol{t}_n)\big)} - \log\kappa\big(\hat{x}(\boldsymbol{t}_n)\big)\bigg].
\end{aligned}
$$

Furthermore, by using the matrix determinant lemma, we can rewrite $\log|\Sigma|$ as

$$
\begin{aligned}
\log|\Sigma| &= \log|\mathcal{H} - \mathcal{H}\mathcal{U}^\top(\boldsymbol{Z} + \mathcal{U}\mathcal{H}\mathcal{U}^\top)^*\mathcal{U}\mathcal{H}| \\
&= \log|\mathcal{H}| - \log|\boldsymbol{Z} + \mathcal{U}\mathcal{H}\mathcal{U}^\top| + \log|(\boldsymbol{Z} + \mathcal{U}\mathcal{H}\mathcal{U}^\top) - (\mathcal{U}\mathcal{H})\mathcal{H}^*(\mathcal{H}\mathcal{U}^\top)| \\
&= \log|\mathcal{H}| - \log|\boldsymbol{Z} + \mathcal{U}\mathcal{H}\mathcal{U}^\top| + \log|\boldsymbol{Z}| \\
&= \log|\mathcal{H}| - \log|\boldsymbol{I}_N + \boldsymbol{Z}^{-1}\boldsymbol{H}|.
\end{aligned}
$$

## S5    Calculation of $h(t, s)$

We apply the *Gelerkin method* [2], a variant of the projection method, to solve equation (S7) with regard to $h(\boldsymbol{t}, \boldsymbol{s})$. Let

$$h(\boldsymbol{t}, \boldsymbol{s}) \simeq \sum_{l=1}^{L} \omega_l \varphi_l(\boldsymbol{t}) \varphi_l(\boldsymbol{s}),$$

and solve for the coefficients, $\{\omega_l\}_{l=1}^{L}$, using the set of residual equations derived from (S7),

$$\iint_{\mathcal{T} \times \mathcal{T}} r(\boldsymbol{t}, \boldsymbol{s}) \varphi_l(\boldsymbol{t}) \varphi_l(\boldsymbol{s}) d\boldsymbol{t} d\boldsymbol{s} = 0, \quad l = 1, \ldots, L,$$

$$r(\boldsymbol{t}, \boldsymbol{s}) \triangleq h(\boldsymbol{t}, \boldsymbol{s}) + \int_{\mathcal{T}} k(\boldsymbol{t}, \boldsymbol{t}') \ddot{\kappa}\big(\hat{x}(\boldsymbol{t}')\big) h(\boldsymbol{t}', \boldsymbol{s}) d\boldsymbol{t}' - k(\boldsymbol{t}, \boldsymbol{s}).$$

Using the relations of the eigenfunctions,

$$\int_{\mathcal{T}} k(\boldsymbol{t}, \boldsymbol{s}) \varphi_l(\boldsymbol{s}) d\boldsymbol{s} = \lambda_l \varphi_l(\boldsymbol{t}), \quad \int_{\mathcal{T}} h(\boldsymbol{t}, \boldsymbol{s}) \varphi_l(\boldsymbol{s}) d\boldsymbol{s} = \omega_l \varphi_l(\boldsymbol{t}),$$

we can solve the residual equations, which results in

$$\omega_l = \frac{\lambda_l}{1 + \lambda_l \Xi_l}, \quad \Xi_l = \int_{\mathcal{T}} \ddot{\kappa}\big(\hat{x}(\boldsymbol{t})\big) \varphi_l^2(\boldsymbol{t}) d\boldsymbol{t}.$$

## S6    Representer Theorem

Formula (S5) provides the MAP estimator $\hat{x}(\boldsymbol{t})$ for general Gaussian Cox processes, but an interesting representation is obtained under a quadratic link function.

If link function $\kappa(x)$ is given in quadratic form,

$$\kappa(x) = x^2, \quad \dot{\kappa}(x) = 2x, \quad \ddot{\kappa}(x) = 2,$$

then (S5) reduces to a Fredholm integral equation of the second kind,

$$\hat{x}(\boldsymbol{t}) + 2 \int_{\mathcal{T}} k(\boldsymbol{t}, \boldsymbol{t}') \hat{x}(\boldsymbol{t}') d\boldsymbol{t}' = \mu + 2 \sum_{n=1}^{N} k(\boldsymbol{t}, \boldsymbol{t}_n) \hat{x}(\boldsymbol{t}_n)^{-1}.$$

The linearity of the integral equation permits a representation of the form

$$\hat{x}(\boldsymbol{t}) - \mu(1 - 2\underline{h}(\boldsymbol{t})) = 2 \sum_{n=1}^{N} h(\boldsymbol{t}, \boldsymbol{t}_n) \hat{x}(\boldsymbol{t}_n)^{-1}, \tag{S8}$$

where $h(\boldsymbol{t}, \boldsymbol{s})$ is the positive semi-definite kernel defined in (S7), and $\underline{h}(\boldsymbol{t}) \triangleq \int_{\mathcal{T}} h(\boldsymbol{t}, \boldsymbol{s}) d\boldsymbol{s}$. (S8) states that MAP estimator $\hat{x}(\boldsymbol{t})$ can be written as expansions in terms of the training examples, where the MAP estimation reduces to a finite-dimensional optimization problem corresponding to new kernel function $h(\boldsymbol{t}, \boldsymbol{s})$. Therefore, the MAP estimator of Gaussian Cox process involves a representer theorem under a quadratic link function, which is a generalization of Wahba's classical representer theorem [12]. Function $h(\boldsymbol{t}, \boldsymbol{s})$ has been studied by [11, 3] as the *equivalent kernel*.

## S7    Kronecker Structure in Product of Eigenfunctions

For a set of $L = \prod_d L_d$ points on a Cartesian product grid (the grid need not be regular), $\boldsymbol{p} \in \mathcal{T}_1 \times \cdots \times \mathcal{T}_D$, matrix $\boldsymbol{\Phi}$, defined by

$$\boldsymbol{\Phi}_{l'l} \triangleq \varphi_{l'}(\boldsymbol{p}_l) = \prod_{d=1}^{D} \varphi_{l'_d}^{(d)}\big(p_{l_d}^{(d)}\big), \quad 1 \le l'_d, l_d \le L_d, \tag{S9}$$

has a Kronecker structure as indicated by

$$\boldsymbol{\Phi} = \boldsymbol{\Phi}^{(1)} \otimes \cdots \otimes \boldsymbol{\Phi}^{(D)}, \quad \boldsymbol{\Phi}_{l'_d l_d}^{(d)} \triangleq \varphi_{l'_d}^{(d)}\big(p_{l_d}^{(d)}\big). \tag{S10}$$

Therefore, in the multi-dimensional input setting, exploiting the Kronecker structure can reduce the $\mathcal{O}(L^2)$ computation in MAP estimation, or $\{\sum_l \lambda_l \beta_l \varphi_l(\boldsymbol{p}_{l'})\}_{l'=1}^{L}$ and $\{\sum_l \beta_l \varphi_l(\boldsymbol{p}_{l'})\}_{l'=1}^{L}$, to $\mathcal{O}(L)$ computation [10], although we did not employ it in this paper.

Table S1: Symbols and Definitions.

| Symbol | Definition |
|---|---|
| $N$ | number of data points |
| $\mathcal{T}$ | observation region |
| $\mathcal{D} = \{\boldsymbol{t}_n \in \mathcal{T}\}_{n=1}^N$ | observed data points |
| $k^{(*)}(\boldsymbol{t}, \boldsymbol{t}')$ | (inverse) kernel function |
| $\mathcal{K}^{(*)} = \int_{\mathcal{T}} k^{(*)}(\boldsymbol{t}, \boldsymbol{t}') \cdot dt'$ | integral operator corresponding to $k^{(*)}(\boldsymbol{t}, \boldsymbol{t}')$ |
| $|\mathcal{K}|$ | functional determinant of operator $\mathcal{K}$ |
| $\int \cdot \mathscr{D}x$ | path integral computation with respect to continuous path $x(\boldsymbol{t})$ |

Table S2: Summary of related works. $\mathcal{O}_M$, $\mathcal{O}_V$, $\mathcal{O}_E$ represent the computational costs of the MAP/predictive mean, the predictive covariance, and the marginal likelihood/evidence lower bound, respectively. $Q$ and $P$ represent the number of gradient descent iterations and the number of MCMC samplings, respectively. For the other symbols, see the main text.

| | Proposed | [13] | [1, 6] | [4] |
|---|---|---|---|---|
| $\mathcal{O}_M$ | $(NL+L^2)Q$ | $(N \cdot \min(N, L))Q$ | $(NL^2+L^3)Q$ | $(NL+L^2)P$ |
| $\mathcal{O}_V$ | $\mathcal{O}_M + N_{\mathrm{mc}}L + (N+L)\cdot\min(N^2, L^2)$ | $\mathcal{O}_M + (N+L)\cdot\min(N^2, L^2)$ | $(NL^2+L^3)Q$ | $(NL+L^2)P$ |
| $\mathcal{O}_E$ | $\mathcal{O}_M + N_{\mathrm{mc}}L + (N+L)\cdot\min(N^2, L^2)$ | $\mathcal{O}_M + (N+L)\cdot\min(N^2, L^2)$ | $(NL^2+L^3)$ | $(NL+L^2)$ |

## S8 Summary of Key Expression and Algorithm

We summarize the key expressions in Section 2.1-2.2 (see Table S1) and the proposed algorithm (see Algorithm 1).

## S9 Summary of Computational Complexity

We provide a summary of the related works about the computational complexity (see Table S2). Here we focus on the algorithms that could scale linearly with the number of data points. $\mathcal{O}_V$ and $\mathcal{O}_E$ of our proposed method reduces to those of [13] under the quadratic link function because the integral in Eq. (19) can be performed analytically. Note that the complexity of each algorithm could be reduced by utilizing a stochastic gradient descent algorithm or by exploiting the Kronecker structure.

## S10 Experimental Settings and Additional Experiments

### S10.1 Experimental Settings

For all the experiments in Section 5, we used a multiplicative Gaussian kernel, $k(\boldsymbol{t}, \boldsymbol{s}) = \prod_d e^{-(\theta(t_d - s_d))^2}$, where hyper-parameter $\theta$ was optimized for each trial by grid search. We calculated the marginal likelihoods or the evidence lower bounds on the following sets of hyper-parameters:

$$\theta_{\lambda_1(t)} = \{.01, .02, \ldots, 0.1\}, \ \theta_{\lambda_2(t)} = \{0.5, 1.0, \ldots, 5.0\}, \ \theta_{\lambda_3(t)} = \{.01, .02, \ldots, 0.1\},$$

for synthetic data, and

$$\theta_{\mathrm{2D\ neuronal\ data}} = \{.01, .02, .03, \ldots, 0.1\}, \ \theta_{\mathrm{3D\ taxi\ data}} = \{0.5, 1.0, 1.5, \ldots, 5.0\},$$

for open real-world data sets. Then we adopted the hyper-parameter that maximized the marginal likelihood or the evidence lower bounds. Note that each of the parameter sets contains values close to those used in [1].

**Algorithm 1** Bayesian Inference via Path Integral Formulation

1: **procedure** INFERENCE($\boldsymbol{t}, \mathcal{D}, \mathcal{T}, J, L, N_{\mathrm{mc}}, N_{\mathrm{gd}}, \kappa(\cdot), k(\cdot, \cdot|\boldsymbol{\theta}), \mu, \{\boldsymbol{p}_l\}_{l=1}^L$)
2: $\quad\{\hat{\beta}_l, \hat{\omega}_l, \lambda_l, \varphi_l(\cdot)\}_{l=1}^L, \log p(\mathcal{D}|\boldsymbol{\theta}) = $ TRAINING($\mathcal{D}, \mathcal{T}, J, L, N_{\mathrm{mc}}, N_{\mathrm{gd}}, \kappa(\cdot), k(\cdot, \cdot|\boldsymbol{\theta}), \mu, \{\boldsymbol{p}_l\}_{l=1}^L$)
3: $\quad$ Predictive mean / MAP: $\hat{x}(\boldsymbol{t}) = \mu + \sum_n k(\boldsymbol{t}, \boldsymbol{t}_n)\gamma\left(\sum_l \hat{\beta}_l \varphi_l(\boldsymbol{t}_n)\right) - \sum_l \lambda_l \hat{\beta}_l \varphi_l(\boldsymbol{t})$
4: $\quad$ Predictive covariance is evaluated by Eq. (14)
5: **procedure** TRAINING($\mathcal{D}, \mathcal{T}, J, L, N_{\mathrm{mc}}, N_{\mathrm{gd}}, \kappa(\cdot), k(\cdot, \cdot|\boldsymbol{\theta}), \mu, \{\boldsymbol{p}_l\}_{l=1}^L$)
6: $\quad\{\lambda_l, \varphi_l(\cdot)\}_{l=1}^L = $ EIGENFUNCTION($\mathcal{T}, J, L, k(\cdot, \cdot|\boldsymbol{\theta})$)
7: $\quad\{\hat{\beta}_l\}_{l=1}^L = $ MAP($\mathcal{D}, L, N_{\mathrm{gd}}, \kappa(\cdot), k(\cdot, \cdot|\boldsymbol{\theta}), \mu, \{\boldsymbol{p}_l\}_{l=1}^L, \{\lambda_l, \varphi_l(\cdot)\}_{l=1}^L$)
8: $\quad\{\hat{\omega}_l\}_{l=1}^L = $ HFUNCTION($\mathcal{T}, L, N_{\mathrm{mc}}, \kappa(\cdot), k(\cdot, \cdot|\boldsymbol{\theta}), \{\hat{\beta}_l, \lambda_l, \varphi_l(\cdot)\}_{l=1}^L$)
9: $\quad\log p(\mathcal{D}|\boldsymbol{\theta}) = $ MARGINALLIKELIHOOD($\mathcal{D}, \mathcal{T}, L, N_{\mathrm{mc}}, \kappa(\cdot), k(\cdot, \cdot|\boldsymbol{\theta}), \mu, \{\hat{\beta}_l, \hat{\omega}_l, \lambda_l, \varphi_l(\cdot)\}_{l=1}^L$)
10: $\quad$**return** $\{\hat{\beta}_l, \hat{\omega}_l, \lambda_l, \varphi_l(\cdot)\}_{l=1}^L,\ \log p(\mathcal{D}|\boldsymbol{\theta})$
11: **procedure** EIGENFUNCTION($\mathcal{T}, J, L, k(\cdot, \cdot|\boldsymbol{\theta})$)
12: $\quad$**for** $d = 1, \ldots, D$ **do**
13: $\quad\quad w = T^{(d)}/J$
14: $\quad\quad$**for** $j, j' = 1, \ldots, J$ **do**
15: $\quad\quad\quad \boldsymbol{K}[j, j'] = k^{(d)}(jw, j'w)$
16: $\quad\quad$Solve $\boldsymbol{K}\boldsymbol{v}_j = e_j \boldsymbol{v}_j \quad : e_1 > e_2 > \cdots > e_J$
17: $\quad\quad \boldsymbol{k}(\cdot) = (k^{(d)}(\cdot, w), k^{(d)}(\cdot, 2w), \ldots, k^{(d)}(\cdot, Jw))^\top$
18: $\quad\quad$**for** $j = 1, \ldots, L_d$ **do**
19: $\quad\quad\quad \lambda_j^{(d)}, \varphi_j^{(d)}(\cdot) = e_j w,\ \boldsymbol{k}(\cdot)^\top \boldsymbol{v}_j/(e_j\sqrt{w})$
20: $\quad U = \varnothing$
21: $\quad$**for** $j_1 = 1, \ldots, L_1$ **do**
22: $\quad\quad \cdots$
23: $\quad\quad$**for** $j_D = 1, \ldots, L_D$ **do**
24: $\quad\quad\quad U = U \cup \left\{\left(\prod_{d=1}^D \lambda_{j_d}^{(d)},\ \prod_{d=1}^D \varphi_{j_d}^{(d)}(\cdot)\right)\right\}$
25: $\quad$**return** $U$
26: **procedure** MAP($\mathcal{D}, L, N_{\mathrm{gd}}, \kappa(\cdot), k(\cdot, \cdot|\boldsymbol{\theta}), \mu, \{\boldsymbol{p}_l\}_{l=1}^L, \{\lambda_l, \varphi_l(\cdot)\}_{l=1}^L$)
27: $\quad$Initialize $\boldsymbol{\beta} \equiv (\beta_1, \ldots, \beta_L)$
28: $\quad$**for** $i = 1, \ldots, N_{\mathrm{gd}}$ **do**
29: $\quad\quad \boldsymbol{\delta} = \nabla_{\boldsymbol{\beta}} \sum_{l=1}^L [r(\boldsymbol{p}_l)]^2 \quad : r(\boldsymbol{p}_l)$ is defined in Eq. (18)
30: $\quad\quad$Update $\boldsymbol{\beta}$ by $Adam(\boldsymbol{\delta})$
31: $\quad$**return** $\boldsymbol{\beta}$
32: **procedure** HFUNCTION($\mathcal{T}, L, N_{\mathrm{mc}}, \kappa(\cdot), k(\cdot, \cdot|\boldsymbol{\theta}), \{\hat{\beta}_l, \lambda_l, \varphi_l(\cdot)\}_{l=1}^L$)
33: $\quad$Sample $N_{\mathrm{mc}}$ points on $\mathcal{T}, \{\boldsymbol{u}_i\}_{i=1}^{N_{\mathrm{mc}}}$
34: $\quad$**for** $l = 1, \ldots, L$ **do**
35: $\quad\quad$Compute $\Xi_l$ by Monte Carlo integration with $\{\boldsymbol{u}_i\}_{i=1}^{N_{\mathrm{mc}}} \quad$ : See Eq. (19)
36: $\quad\quad \omega_l = \lambda_l/(1 + \lambda_l \Xi_l)$
37: $\quad$**return** $\{\omega_l\}_{l=1}^L$
38: **procedure** MARGINALLIKELIHOOD($\mathcal{D}, \mathcal{T}, L, N_{\mathrm{mc}}, \kappa(\cdot), k(\cdot, \cdot|\boldsymbol{\theta}), \mu, \{\hat{\beta}_l, \hat{\omega}_l, \lambda_l, \varphi_l(\cdot)\}_{l=1}^L$)
39: $\quad$Sample $N_{\mathrm{mc}}$ points on $\mathcal{T}, \{\boldsymbol{u}_i\}_{i=1}^{N_{\mathrm{mc}}}$
40: $\quad$Compute $\log p(\mathcal{D}|\boldsymbol{\theta})$ by Monte Carlo integration with $\{\boldsymbol{u}_i\}_{i=1}^{N_{\mathrm{mc}}} \quad$ : See Eq. (20)
41: $\quad$**return** $\log p(\mathcal{D}|\boldsymbol{\theta})$

Table S3: Results on three types of synthetic data with standard errors in brackets.

| $\lambda_1(t)$ | $L=3$ | | | $L=5$ | | | $L=10$ | | | $L=20$ | | |
|---|---|---|---|---|---|---|---|---|---|---|---|---|
| | IQL$_{.5}$ | IQL$_{.85}$ | Time | IQL$_{.5}$ | IQL$_{.85}$ | Time | IQL$_{.5}$ | IQL$_{.85}$ | Time | IQL$_{.5}$ | IQL$_{.85}$ | Time |
| PIF$_s$ | 13.00 | 9.68 | 10.33 | 11.33 | 7.64 | 10.19 | 11.56 | 8.02 | 10.49 | 11.58 | 7.62 | 10.96 |
| | (2.24) | (3.48) | (0.37) | (3.10) | (3.44) | (0.39) | (3.43) | (3.38) | (0.47) | (3.26) | (3.22) | (0.75) |
| PIF$_e$ | 12.65 | 9.56 | 10.09 | 12.04 | 9.26 | 10.01 | 11.89 | 9.50 | 10.11 | 12.29 | 9.30 | 10.45 |
| | (1.76) | (2.06) | (0.37) | (3.38) | (2.95) | (0.31) | (4.43) | (3.70) | (0.30) | (4.33) | (3.74) | (0.40) |
| PIF$_q$ | 11.80 | 8.65 | 9.91 | 11.88 | 8.17 | 9.86 | 12.73 | 9.02 | 10.05 | 12.50 | 9.00 | 10.49 |
| | (2.19) | (2.68) | (0.24) | (3.15) | (2.93) | (0.30) | (4.30) | (3.92) | (0.31) | (4.40) | (4.29) | (0.58) |
| STVB | 12.59 | 8.70 | 34.86 | 11.81 | 8.20 | 34.88 | 11.86 | 8.39 | 35.25 | 12.04 | 8.54 | 35.90 |
| | (2.17) | (2.91) | (1.15) | (2.45) | (2.97) | (0.99) | (2.82) | (3.31) | (1.13) | (2.99) | (3.58) | (1.21) |
| VBPP | 12.12 | 8.65 | 24.69 | 12.18 | 8.44 | 25.12 | 11.69 | 7.68 | 26.31 | 15.99 | 10.92 | 27.38 |
| | (1.99) | (2.35) | (0.92) | (3.61) | (3.39) | (1.00) | (4.94) | (4.06) | (1.16) | (3.17) | (4.77) | (0.79) |

| $\lambda_2(t)$ | $L=3$ | | | $L=5$ | | | $L=10$ | | | $L=20$ | | |
|---|---|---|---|---|---|---|---|---|---|---|---|---|
| | IQL$_{.5}$ | IQL$_{.85}$ | Time | IQL$_{.5}$ | IQL$_{.85}$ | Time | IQL$_{.5}$ | IQL$_{.85}$ | Time | IQL$_{.5}$ | IQL$_{.85}$ | Time |
| PIF$_s$ | 15.57 | 15.38 | 10.02 | 15.05 | 10.66 | 9.86 | 14.57 | 11.10 | 10.44 | 14.46 | 10.96 | 10.51 |
| | (1.64) | (4.92) | (0.22) | (1.06) | (3.89) | (0.15) | (0.93) | (3.53) | (0.73) | (0.83) | (3.20) | (0.50) |
| PIF$_e$ | 34.73 | 30.96 | 9.81 | 19.41 | 13.83 | 9.62 | 15.95 | 12.56 | 9.99 | 15.68 | 12.35 | 10.12 |
| | (13.79) | (8.88) | (0.28) | (8.76) | (5.81) | (0.21) | (2.24) | (2.71) | (0.51) | (1.67) | (1.99) | (0.40) |
| PIF$_q$ | 21.83 | 29.12 | 9.64 | 15.23 | 11.31 | 9.57 | 14.46 | 10.02 | 10.04 | 13.05 | 8.65 | 10.15 |
| | (3.71) | (8.91) | (0.17) | (2.98) | (3.06) | (0.15) | (1.96) | (2.61) | (0.64) | (1.88) | (1.70) | (0.38) |
| STVB | 15.37 | 9.51 | 34.47 | 14.71 | 10.24 | 34.45 | 14.48 | 10.03 | 36.10 | 13.38 | 9.10 | 36.18 |
| | (0.96) | (2.62) | (0.35) | (1.41) | (2.61) | (0.22) | (1.48) | (2.72) | (1.90) | (1.91) | (2.69) | (1.17) |
| VBPP | 29.54 | 15.92 | 23.80 | 15.18 | 10.20 | 23.90 | 14.79 | 10.14 | 26.66 | 13.13 | 8.79 | 26.66 |
| | (4.17) | (0.83) | (0.36) | (1.67) | (1.43) | (0.29) | (1.77) | (2.07) | (1.67) | (1.69) | (1.45) | (0.79) |

| $\lambda_3(t)$ | $L=3$ | | | $L=5$ | | | $L=10$ | | | $L=20$ | | |
|---|---|---|---|---|---|---|---|---|---|---|---|---|
| | IQL$_{.5}$ | IQL$_{.85}$ | Time | IQL$_{.5}$ | IQL$_{.85}$ | Time | IQL$_{.5}$ | IQL$_{.85}$ | Time | IQL$_{.5}$ | IQL$_{.85}$ | Time |
| PIF$_s$ | 45.17 | 35.01 | 11.05 | 32.59 | 20.21 | 10.80 | 25.60 | 14.75 | 10.87 | 27.10 | 15.81 | 10.90 |
| | (4.26) | (9.64) | (0.33) | (7.24) | (6.94) | (0.29) | (6.17) | (3.80) | (0.27) | (6.12) | (3.06) | (0.32) |
| PIF$_e$ | 39.69 | 26.70 | 10.24 | 32.24 | 21.05 | 10.17 | 31.46 | 21.89 | 10.53 | 31.23 | 21.15 | 10.48 |
| | (4.55) | (6.15) | (0.27) | (7.14) | (4.79) | (0.39) | (6.17) | (3.92) | (0.44) | (5.83) | (4.20) | (0.32) |
| PIF$_q$ | 39.82 | 25.42 | 10.47 | 30.81 | 20.03 | 10.42 | 29.89 | 18.73 | 10.41 | 31.36 | 19.96 | 10.46 |
| | (4.89) | (5.88) | (0.21) | (5.89) | (4.12) | (0.29) | (4.47) | (3.53) | (0.27) | (5.91) | (3.56) | (0.32) |
| STVB | 42.41 | 23.01 | 40.02 | 30.89 | 16.75 | 48.83 | 28.87 | 17.37 | 38.98 | 28.59 | 16.75 | 38.35 |
| | (5.71) | (5.95) | (1.20) | (7.04) | (3.17) | (37.64) | (6.76) | (3.31) | (0.65) | (6.46) | (3.46) | (1.04) |
| VBPP | 38.73 | 22.79 | 26.60 | 30.97 | 19.41 | 25.88 | 30.50 | 19.31 | 26.86 | 37.09 | 25.06 | 28.14 |
| | (4.35) | (4.14) | (0.74) | (4.88) | (3.23) | (0.70) | (5.37) | (3.67) | (0.84) | (6.90) | (3.57) | (0.90) |

For fair comparison, we employed a popular (batch) gradient descent algorithm, *Adam* [5], to perform estimations for all compared methods. We equally set the number of iteration as 5,000, but used different learning parameters for the models: 0.5 for PIF$_e$; 0.05 for PIF$_q$; 0.05 for PIF$_s$; 0.005 for STVB; 0.05 for VBPP. We implemented all the models by using TensorFlow-2.2, where for STVB we used the python code provided by Aglietti et al. [1]. Each of the CPU times reported is the amount of time required to calculate the MAP/predictive mean, the predictive covariance, and the marginal likelihood given a hyper-parameter, where computing the eigenfunctions of the kernel is of course included in the CPU time.

## S10.2 Figure 1A in tabular form

We provide the results in Figure 1A in tabular form (see Table S3) to make the results easy to review.

## S10.3 Experiments of How Stably the Proposed Scheme Works on Real-world Data

To demonstrate that our scheme is stable on real-world data, we ran additional experiments based on the real-world data used in Figure 2. For the 2D neuronal data ($N_{\text{train}} = 583$, $N_{\text{test}} = 29127$), we extracted two training datasets of 583 data points from the original test data randomly, which resulted in three training ($N_{\text{train}} = 583$) and a test ($N_{\text{test}} = 27961$) datasets. For the 3D taxi data ($N_{\text{train}}$

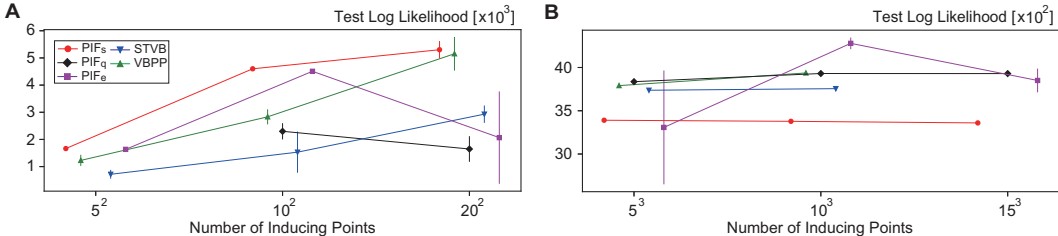

Figure S1: The predictive performances on open real-world data. The error bars represent the standard deviations. (A) 2D neuronal data. (B) 3D spatio-temporal taxi data.

$= 1000$, $N_{\text{test}} = 3401$), we extracted two training datasets of 1000 data points from the original test data randomly, which resulted in three training ($N_{\text{train}} = 1000$) and a test ($N_{\text{test}} = 1401$) datasets. We evaluated three times the predictive performances of the compared models in terms of the test log likelihood, and calculated the means and the standard deviations (error bars) of the performances. Figure S1 shows the results.

## S10.4   Experiments on Larger Taxi Dataset

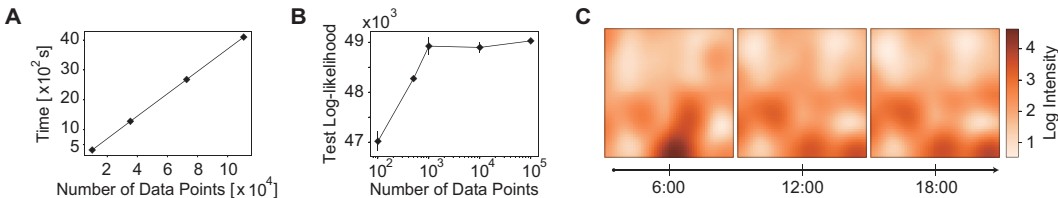

Figure S2: Results on large 3D taxi datasets. (A) The CPU times. (B) The test log-likelihoods. The error bars represent the standard deviations. (C) The estimated intensity function on the dataset with 110,705 data points.

To demonstrate that our scheme is scalable to large data size, we ran an additional experiment on a larger 3D taxi dataset [7]. The area considered was the same as that used in Experiments, and we took 10, 50, 100, and 150 weekdays from 1 July 2013, resulting in the training data sets containing from 9,971 to 110,705 data points. It should be noted that the maximum data size is comparable to that considered in [4] (113,020). We employed $\text{PIF}_{\text{q}}$ with $L = 10^3$, and plot the execution times with respect to the number of data points (see Figure S2A). Also, we randomly divided the datasets ($N = 110,705$) into the training ($N = 100,000$) and the test ($N = 10,705$) data, ran $\text{PIF}_{\text{q}}$ on various sizes of subsets of the training data, and evaluated the test log-likelihoods. Figure S2B plots the test log-likelihoods as functions of the number of data points used for training; it shows that our approach ($\text{PIF}_{\text{q}}$) can process large datasets effectively and recover the underlying intensity function more accurately with larger training datasets.

## S10.5   Application of Stochastic Optimization Algorithm

Because the objective function to be minimized in the MAP estimation, $\sum_{l=1}^{L}[r(\boldsymbol{p}_l)]^2$, is given as a sum of residuals, we can apply a mini-batch gradient descent (MGD) algorithm to the optimization problem. With mini-batch size $\tilde{L} \ll L$, MGD reduces the computational complexity for each iteration to $\mathcal{O}(NL + N\tilde{L} + \tilde{L}^2) \simeq \mathcal{O}(NL)$, where the dominant cost stems from $\{\gamma[\sum_l \beta_l \varphi_l(\boldsymbol{t}_n)]\}_{n=1}^{N}$. Figure S3 shows the performances of MGD with $\tilde{L} = 258$ and the batch gradient descent (BGD) on the 3D spatio-temporal taxi data used in Section 5 ($N_{\text{train}} = 1000$), where the number of inducing points, $L$, was set as $20^3$ for all models. In $\text{PIF}_{\text{q}}$ and $\text{PIF}_{\text{e}}$, MGD (red line) converged much faster than BGD (blue line), which suggests that MGD would enhance the practical utility of our scheme under a large number of inducing points. In $\text{PIF}_{\text{s}}$, however, BGD achieved the convergence so rapidly and MGD worked poorly compared to BGD. Here we used the learning parameters of

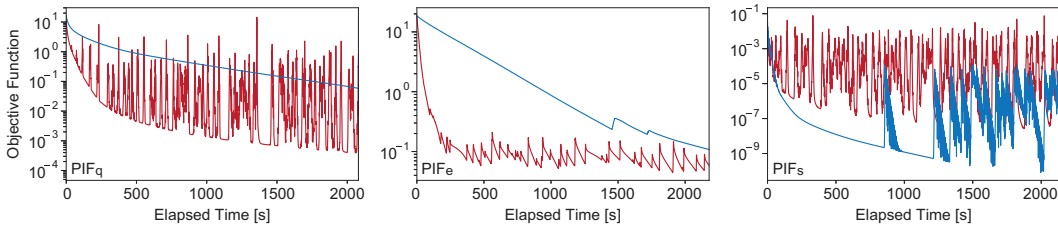

Figure S3: Training of MAP estimator on 3D spatio-temporal taxi data. $L$ was set to $20^3$ for each model. The blue and red lines represent the results yielded by batch gradient descent and mini-batch gradient descent with mini-batch size of 258, respectively. The maximum of the x-axis (elapsed time) in each figure equals the elapsed time that the batch gradient descent needed to execute the 5,000 epoch updates.

$10^{-3}$, $10^{-2}$, and $10^{-5}$ for $\text{PIF}_q$, $\text{PIF}_e$, and $\text{PIF}_s$, respectively, but examining the parameters of Adam [5] more carefully would speed up the convergence.