# OpenReview forum: "Fast Bayesian Inference for Gaussian Cox Processes via Path Integral Formulation"
_NeurIPS.cc/2021/Conference — NeurIPS 2021 Spotlight_

### Official Review · Reviewer_SdDJ · 2021-07-12

**Rating:** 7
**Confidence:** 4

**Summary:**

The paper addresses the inference of Gaussian Cox processes (GCP), and provides the path integral formulation to derive a MAP estimate and a Laplace approximation for Bayesian posterior inference. The strongest contribution is, that the formulation holds for general GP prior kernels, and for a variety of link functions. Comparison of the new methods is done on simulated and real datasets.

**Limitations And Societal Impact:**

For limitations see above. Societal impact is briefly discussed at the end of the manuscript.

**Main Review:**

I found the paper nicely written, and the derivations elegant, and consider them as a good contribution to the active research problem of GCP inference. The formulation allows usage of various link functions and kernels. However, when it comes to the algorithm, I feel that the authors are not discussing all practical challenges.

+ A major point is the optimization of hyperparameters. I know, that is always an ugly task, but it is important for practical applications, and not discussed at all in the manuscript (at least I didn’t find it). I have the feeling, that it is difficult with the proposed task, because you would need to recompute the eigenfunctions of the kernel, every time hyperparameters are updated. That is potentially the reason, why the authors resorted to grid search. However, other approaches allow for optimization with gradient methods (e.g. the VBPP). Also, it is not clear to me, whether the grid search is included in the CPU times, that are reported.
+ I like the complexity analysis, but I miss the discussion of the complexity of finding the eigenfunctions of the kernel operator. Though, this has to be done only the beginning (as long as kernel is fixed), I think, it should be discussed here. Also, that for finding the basis, D one dimensional integrals have to solved numerically, should be part of the story. In general, I’d appreciate, if the authors could include a bit more details about finding the eigenfunctions in the main text.

## Update

I thank the authors for their response, and I raised my score given the promised changes of the manuscript. I think the final manuscript will be a strong publication.
I believe the paper is interesting, and I don’t consider any of the points above as a deathblow to the paper, but I would like to see them discussed.

Some minors:
+ Would be interesting to see comparison also with [24] and [13] in the comparisons, since the first has same complexity as the presented method and the latter is a very related method, as the authors state themselves. If code is available, that is.
+ I didn’t like that t is used for the space variable and transpose operation. Maybe use \dagger or \top for the latter.
+ L.125: an action integral


**Time Spent Reviewing:**

5h

---

> ### Author Response · Authors · 2021-08-10
> **To Reviewer SdDJ**
>
> We would like to thank the reviewer for poining out our insufficient exlanations and providing an important perspective to discuss in the paper. Below we provide a detailed response to each of the major and minor comments, which we believe is satisfactory.
>
>
>
> **About the optimization of the hyperparameters and CPU time:**
>
> For all the compared methods, we performed hyperparameter optimization by means of the same strategy, that is, grid search: we calculated the marginal likelihood or evidence lower bound for each of the values of hyperparameters shown in the supplementary material (S8.1), and adopted the hyperparameter that maximized the marginal likelihood or evidence lower bound. Each of the CPU times reported is the amount of time required to calculate the MAP/predictive mean, the predictive covariance, and the marginal likelihood given a hyperparameter, where computing the eigenfunctions of the kernel is of course included in the CPU time. We will state the definition of the CPU time in the manuscript.
>
> We did not consider optimizing hyperparameters with gradient descent algorithms in this paper, but as the reviewer pointed out, VB-based approaches allow for optimization with gradient methods, which are particularly beneficial when the kernel function has several kinds of hyperparameters. It should be emphasized here that our approach can also employ gradient methods: The hyperparameter optimization in our approach, where the objective function $L(\theta, \hat{\beta}(\theta)) \equiv \log p(\mathcal{D}|\theta)$ has an argmin $\hat{\beta} (\theta) \equiv \\{ \\hat{\beta}_ l \\}_{l=1}^L $ (see Eq. (18)), belongs to the bi-level optimization, and the exact computation of the gradient of $L(\theta, \hat{\beta}(\theta))$ can be executed (Gould et al., "On differentiating parameterized argmin and argmax problems with application to bi-level optimization", arXiv, 2016); Here we show only the result,
> \begin{equation}
> \frac{d L(\theta, \hat{\beta}(\theta))}{d\theta} = \frac{\partial L\}{\partial \theta} - \biggl( \nabla _ {\beta} L(\theta,\hat{\beta}) \biggr)^{\top} \biggl( \nabla^2 _ {\beta \beta} f(\theta, \hat{\beta}) \biggr)^{-1} \biggl( \frac{\partial}{\partial \theta} \nabla _ {\beta} f(\theta, \hat{\beta}) \biggr), \quad f(\theta, \hat{\beta}) \equiv \sum _ {l=1}^L [r(p_l)]^2.
> \end{equation}
> When automatic differentiation is employed, the computational complexity of computing the gradient $\frac{d L(\theta, \hat{\beta}(\theta))}{d\theta}$ is equal to the sum of $\mathcal{O}(L^3)$ and the complexity of the marginal likelihood (see Table A in response to Reviewer rjT5), where $\mathcal{O}(L^3)$ comes from the inversion of matrix, $\nabla^2 _ {\beta \beta} f(\theta, \hat{\beta})$. When a gradient descent algorithm is employed for hyperparameter optimization, our approach needs to alternate between the optimization in the $L$-dimensional space (MAP estimation) and that in the dim($\theta$)-dimensional space, while VB-based approaches perform the optimization in higher $(L^2 + \text{dim}(\theta))$-dimensional space. Although it is out-of-scope of this paper, a comparative analysis between our approach and VB-based alternatives to investigate the practical utility of gradient descent algorithms is an important next step for future work. We will add the discussion so far in the main text.
>
>
>
> **The discussion of the complexity of finding the eigenfunctions of the kernel operator:**
>
> Under the Nystrom method with $J$-point numerical integration, the integral equation to obtain eigenfunctions of the kernel operator reduces to an eigen equation of a $J \times J$ matrix (for details, see S7). Therefore, the computational complexity of finding the eigenfunctions is $\mathcal{O}(J^3)$, and when a multiplicative kernel function is assumed on the $D$-dimensional setting, the computational complexity is $\mathcal{O}(DJ^3)$. We set $J = 1000$ for each of the dimensions in the paper, which is negligible compared to the cost of the MAP estimation. It is worth noting that the gradients of the obtained eigenfunctions and eigenvalues with respect to the hyperparameter can be evaluated analytically, which allows us to obtain the gradient of the marginal likelihood as discussed above. According to the reviewer's suggestion, we will include how to find eigenfunctions and these discussions in the main text.
>
>
>
> **Would be interesting to see comparison also with [24] and [13]:**
>
> This time it is difficult to perform the comparison with [24] and [13] because the codes are not available. We hope that it would be acceptable for the reviewer.
>
>
>
> **Symbol of transpose operation:**
>
> Following the reviewer suggestion, We will use $\top$ for transpose operation.

---

### Official Review · Reviewer_SxYf · 2021-07-15

**Rating:** 5
**Confidence:** 3

**Summary:**

   The manuscript tackles the problem of scalable and tractable approximate inference for Gaussian Cox models using a path integral approach. Experiments on synthetic 1D data and on 2D neural data illustrate certain properties of the approach.

   Strengths
   + Gaussian Cox processes are a relevant class of Bayesian models that are computationally intractable and hence a meaningful research objective.


   Weaknesses
   - The paper does not come with code; hence it will be hard to reproduce the results.
   - The data sets considered in the main paper are rather small compared to [24]. A direct comparison to [24] in the main paper would have been instructive.
   - The empirical evaluation does not support a solid claim on the runtime and memory footprint of the method.
   - An empirical comparison to previous methods e.g. [24] is only partly available.

**Ethical Concerns:**

No concerns.

**Limitations And Societal Impact:**

Yes.

**Main Review:**

   1) Clarity

   The technical content of the paper on page 3 in particular which is concluded by the strange sentence in lines 117-118 is hard to grasp as the authors chose to use an "intuitive" way of deriving the results. A concise summary of the key expressions and the relevant computations e.g. in an algorithm box would have been beneficial.

   2) Originality

   The use of the path integral methodology seems not explored beforehand.

   3) Significance

   The method is specific to the Gaussian Cox process and the benefits w.r.t. to previous methods in terms of theory and empirical performance are not fully established. Hence, the significance of the work is limited.

   4) Reproducibility

   As the code to run the experiments is not included in the submission, it should be rather difficult to reproduce the results.

   5) Empirical analysis

   The manuscript contains experiments on synthetic and real-world data. But the evidence backing up the "fast" from the title is limited. It is hard to assess how accurate the approximation is and how it compares to other competing methods e.g. [24]. An experiment backing up the claim in the abstract on the benefits in the "multidimensional input setting" where "the number of inducing inputs tends to be large" should be added.

   6) Minor Issues and Typos

   line 3: In this paper, we propose
   line 49: inferencing, lines 80 and 92: of GP prior, line 133: is that it can
   Figure 2: over10
   Capitalisation in References: Integralgleichungen, Anwendungen

**Time Spent Reviewing:**

3

---

> ### Author Response · Authors · 2021-08-10
> **To Reviewer SxYf**
>
> We would like to thank the reviewer for the careful reading and constructive comments. Below we provide a detailed response to each of the major comments.
>
>
> **Concern about clarity:**
>
> According to the reviewer's suggestion, we will add a concise summary of the key expressions for Section 2.1-2.2. as follows.
>
> | Symbol                                                       | Definition                                                   |
> | ------------------------------------------------------------ | ------------------------------------------------------------ |
> | $N$                                                          | number of data points                                        |
> | $\mathcal{T}$                                                | observation region                                           |
> | $\mathcal{D}=\\{t_n\in \mathcal{T} \\}_{n=1}^N$              | observed data points                                         |
> | $k^{(*)}(t,t')$                                              | (inverse) kernel function                                    |
> | $\mathcal{K}^{(\*)} = \int_{\mathcal{T}} k^{(\*)}(t,t')\cdot dt'$ | integral operator corresponding to $k^{(*)}(t,t')$           |
> | $\|\mathcal{K}\|$                                            | functional determinant of operator $\mathcal{K}$             |
> | $\int \cdot \mathscr{D}x$                                    | path integral computation with respect to continuous path $x(t)$ |
>
> &nbsp;&nbsp;
>
> **Concern about lack of reproducibility:**
>
> The reviewer is concerned about lack of reproducibility, but doesn’t clarify what is inadequate. We think that we have provided enough details to reproduce our results, but if something is missing, please point out it, which we will provide. Unfortunately, we cannot share our code due to the regulations of the organization we belong to, but we believe that not including the code itself does not lead to the lack of reproducibility.
>
>
>
> **It is hard to assess how accurate the approximation is in empirical analysis:**
>
> In Experiments (Section 5), we compared our method with alternative methods based on several predictive performance measures, where the conditions, which includes the shape of kernel function, were aligned among the methods as much as possible. Then we obtained a conclusion that our method achieved the comparable accuracy against the VB-based alternatives. Especially, the comparison between PIF$_q$ and VBPP is informative because the both used the same kernel and link functions but relied on different approximations. VB-based approximations are usually better at estimating posterior distributions than Laplace approximations, but Figure 1 shows that our PIF$ _q$ with Laplace approximation performed comparably against VBPP under $L \gtrsim 5$ with regard to both IQL.85 and IQL.50, which demonstrates the practical utility for recovering the posterior distribution as well as point estimation. We will add the discussion about approximation algorithms to the main text.
>
>
>
> **It is hard to assess how it compares to other competing methods e.g. [24]:**
>
> It could be possible to make a fair comparison between our method and the MCMC-based method [24] by adopting the quadratic link function and Matern kernel function in our method, but this time it is hard to implement the experiment because the code of [24] is not available. We will make a clear statement in the paper that our focus is on a deterministic approach with the most standard Gaussian kernel function, which makes our position clearer. Also, we will state in Discussion that it is an important next step to make a comparative analysis with the scalable MCMC-based method.
>
>
>
> **An experiment backing up the claim in the abstract on the benefits in the "multidimensional input setting" where "the number of inducing inputs tends to be large" should be added:**
>
> In Section 5.2, we performed an experiment based on a spatio-temporal data, that is, three-dimensional data, where the number of inducing points tends to be large (e.g. $10^3$). Figure 2B shows that our approach achieved comparable predictive accuracy while being faster than the VB-based methods, which we believe is consistent with the claim in the abstract. We did not consider more high-dimensional scenarios since point processes are utilized to analyze at most three-dimensional data in practice.

---

> > ### Comment · Reviewer_SxYf · 2021-08-22
> > **Reply to authors**
> >
> > Thanks for your reply.
> >
> > I still think that not sharing the code impedes reprodicibility. You do not need to share the entire code base but a valid and instructive demo would already be good step. Your statement that a comparison to [24] is hard exactly points at the problem. Hence, your paper comes without a direct comparison which makes the empirical part only partly convincing.
> >
> > I might have missed some important theoretical point to judge the merit of the paper, so I'm lowering my confidence and increase my score.

---

### Official Review · Reviewer_jgS8 · 2021-07-16

**Rating:** 7
**Confidence:** 2

**Summary:**

This paper gives a path integral formulation of a Gaussian Cox process (GCP) with an accompanying inference framework for efficiently resolving the integrals. The proposed method is empirically validated on a series of synthetic and real-world data examples where the results seem promising.


**Limitations And Societal Impact:**

The authors do highlight two limitations of this work.

**Main Review:**

I found this article to be an enjoyable read. The authors do a very good job of motivating the topic before presenting the proposed model in an intuitive step-by-step framework. There is a lengthy related work discussion that effectively places the proposed model within the GCP literature. Finally, the experimental results are compelling with the proposed path-integral GCP obtaining comparable performance to existing methods, but with substantial computational accelerations. It is of note that some of the concepts in this paper were new to me, however, the accompanying appendix does a good job of providing background information for readers like myself. I have some minor points that I will outline below.

Main comments:

1. Are you able to provide error bars in Figure 2? It would be interesting to see how stable each of the methods is on real-world data.
2. What is the memory footprint of your method in comparison to STVB and VBPP methods? Given the method's computational runtime performance, this result would be of interest.
3. Could you clarify what is meant in lines 94-96 of the Appendix? You mention that the kernel's hyperparameters are learned through optimisation of the marginal likelihood, but then write that the same hyperparameter is learned through grid search which confuses me.
4. I am interested to see how this approach scales to large data sets. I can see that you have considered an example with up to 110705 points in Section 8.3 of the appendix but you only report timings. Do you have any metrics to show the quality of your model here, such as test log-likelihood, as I would be interested to see if the model's capacity to recover the underlying process is maintained with such large datasets?


Minor comments:

1. Can you explain the behaviour of PIFq in Figure 2? The model seems unstable
2. Typo in the caption of Figure 2 - "over10 hours".
3. Given the similarity of all five models' behaviour in Figure 1, I found the plots quite hard to review. My opinion is that these results may be better suited in a table.
4. I agree with the authors that providing a measure-theoretic formulation of the path integral would be a useful next step for future work in this area.

**Time Spent Reviewing:**

5

---

> ### Author Response · Authors · 2021-08-10
> **To Reviewer jgS8**
>
> We would like to thank the reviewer for the positive and constructive comments. Below we provide a detailed response to each of the major and minor comments.
>
>
>
> **Error bars in Figure 2:**
>
> By dividing the training dataset into three subsets, we can calculate the mean and the standard deviation (error bar) of the predictive performance in Figure 2. Unfortunately, we cannot finish the task by the deadline of the author response. We promise to add a figure of predictive performances with error bars to the supplementary material, if our paper is accepted.
>
>
>
> **Memory footprint of the proposed method:**
>
> The memory complexity of our proposed method is $\mathcal{O}(NL + L^2)$, which is the same as that of STVB or VBPP. We will add the explanation to the main text.
>
>
>
> **About the optimization of the hyperparameters:**
>
> We apologize for our confusing explanation. In the paper, we performed hyperparameter optimization by grid search: we calculated the marginal likelihood for each of the values of hyperparameters shown in the supplementary material (S8.1), and adopted the hyperparameter that maximized the marginal likelihood. We will modify the lines 94-96 in the supplementary material as above to avoid any misunderstandings. About related discussions, please see our responses to Reviewer SdDJ.
>
>
>
> **The model's capacity to recover the underlying process on larger taxi dataset:**
>
> According to the reviewer's suggestion, we ran an additional experiment based on the test log-likelihood: We divided the datasets ($N = 110,705$) into the training ($N = 100,000$) and the test ($N = 10,705$) data randomly, ran PIF$_ q$ on various sizes of subsets of the training data, and evaluated the test log-likelihoods. Table B displays the test log-likelihoods $LL$ as functions of the data size $N$ used for training, showing that our approach (PIF$_q$) can eat large datasets effectively and recover the underlying intensity function more accurately on larger training datasets. We will add the result to the supplementary material.
>
> | $N$  |    100     |    500    |   1,000    |   10,000   | 100,000 |
> | :--: | :--------: | :-------: | :--------: | :--------: | :-----: |
> | $LL$ | 47016(197) | 48272(40) | 48922(179) | 48896(103) |  49027  |
>
> **Table B:** The test log-likelihoods on large 3D taxi dataset with standard errors in brackets.
>
> &nbsp;
>
> **Quite hard to review the results in Figure 2:**
>
> According to the reviewer's suggestion, we will add the following table.
>
> $\lambda_1(t)$
>
> |         |  $L = 3$   |             |        |  $L = 5$   |             |        |  $L = 10$  |             |        |  $L = 20$  |             |        |
> | ------- | :--------: | :---------: | :----: | :--------: | :---------: | :----: | :--------: | :---------: | :----: | :--------: | :---------: | :----: |
> |         | IQL$_{.5}$ | IQL$_{.85}$ |  Time  | IQL$_{.5}$ | IQL$_{.85}$ |  Time  | IQL$_{.5}$ | IQL$_{.85}$ |  Time  | IQL$_{.5}$ | IQL$_{.85}$ |  Time  |
> | PIF$_s$ |   13.00    |    9.68     | 10.33  |   11.33    |    7.64     | 10.19  |   11.56    |    8.02     | 10.49  |   11.58    |    7.62     | 10.96  |
> |         |   (2.24)   |   (3.48)    | (0.37) |   (3.10)   |   (3.44)    | (0.39) |   (3.43)   |   (3.38)    | (0.47) |   (3.26)   |   (3.22)    | (0.75) |
> | PIF$_e$ |   12.65    |    9.56     | 10.09  |   12.04    |    9.26     | 10.01  |   11.89    |    9.50     | 10.11  |   12.29    |    9.30     | 10.45  |
> |         |   (1.76)   |   (2.06)    | (0.37) |   (3.38)   |   (2.95)    | (0.31) |   (4.43)   |   (3.70)    | (0.30) |   (4.33)   |   (3.74)    | (0.40) |
> | PIF$_q$ |   11.80    |    8.65     |  9.91  |   11.88    |    8.17     |  9.86  |   12.73    |    9.02     | 10.05  |   12.50    |    9.00     | 10.49  |
> |         |   (2.19)   |   (2.68)    | (0.24) |   (3.15)   |   (2.93)    | (0.30) |   (4.30)   |   (3.92)    | (0.31) |   (4.40)   |   (4.29)    | (0.58) |
> | STVB    |   12.59    |    8.70     | 34.86  |   11.81    |    8.20     | 34.88  |   11.86    |    8.39     | 35.25  |   12.04    |    8.54     | 35.90  |
> |         |   (2.17)   |   (2.91)    | (1.15) |   (2.45)   |   (2.97)    | (0.99) |   (2.82)   |   (3.31)    | (1.13) |   (2.99)   |   (3.58)    | (1.21) |
> | VBPP    |   12.12    |    8.65     | 24.69  |   12.18    |    8.44     | 25.12  |   11.69    |    7.68     | 26.31  |   15.99    |    10.92    | 27.38  |
> |         |   (1.99)   |   (2.35)    | (0.92) |   (3.61)   |   (3.39)    | (1.00) |   (4.94)   |   (4.06)    | (1.16) |   (3.17)   |   (4.77)    | (0.79) |
>
> $\lambda_2(t)$
>
> |         |  $L = 3$   |             |        |  $L = 5$   |             |        |  $L = 10$  |             |        |  $L = 20$  |             |        |
> | ------- | :--------: | :---------: | :----: | :--------: | :---------: | :----: | :--------: | :---------: | :----: | :--------: | :---------: | :----: |
> |         | IQL$_{.5}$ | IQL$_{.85}$ |  Time  | IQL$_{.5}$ | IQL$_{.85}$ |  Time  | IQL$_{.5}$ | IQL$_{.85}$ |  Time  | IQL$_{.5}$ | IQL$_{.85}$ |  Time  |
> | PIF$_s$ |   15.57    |    15.38    | 10.02  |   15.05    |    10.66    |  9.86  |   14.57    |    11.10    | 10.44  |   14.46    |    10.96    | 10.51  |
> |         |   (1.64)   |   (4.92)    | (0.22) |   (1.06)   |   (3.89)    | (0.15) |   (0.93)   |   (3.53)    | (0.73) |   (0.83)   |   (3.20)    | (0.50) |
> | PIF$_e$ |   34.73    |    30.96    |  9.81  |   19.41    |    13.83    |  9.62  |   15.95    |    12.56    |  9.99  |   15.68    |    12.35    | 10.12  |
> |         |  (13.79)   |   (8.88)    | (0.28) |   (8.76)   |   (5.81)    | (0.21) |   (2.24)   |   (2.71)    | (0.51) |   (1.67)   |   (1.99)    | (0.40) |
> | PIF$_q$ |   21.83    |    29.12    |  9.64  |   15.23    |    11.31    |  9.57  |   14.46    |    10.02    | 10.04  |   13.05    |    8.65     | 10.15  |
> |         |   (3.71)   |   (8.91)    | (0.17) |   (2.98)   |   (3.06)    | (0.15) |   (1.96)   |   (2.61)    | (0.64) |   (1.88)   |   (1.70)    | (0.38) |
> | STVB    |   15.37    |    9.51     | 34.47  |   14.71    |    10.24    | 34.45  |   14.48    |    10.03    | 36.10  |   13.38    |    9.10     | 36.18  |
> |         |   (0.96)   |   (2.62)    | (0.35) |   (1.41)   |   (2.61)    | (0.22) |   (1.48)   |   (2.72)    | (1.90) |   (1.91)   |   (2.69)    | (1.17) |
> | VBPP    |   29.54    |    15.92    | 23.80  |   15.18    |    10.20    | 23.90  |   14.79    |    10.14    | 26.66  |   13.13    |    8.79     | 26.66  |
> |         |   (4.17)   |   (0.83)    | (0.36) |   (1.67)   |   (1.43)    | (0.29) |   (1.77)   |   (2.07)    | (1.67) |   (1.69)   |   (1.45)    | (0.79) |
>
> $\lambda_3(t)$
>
> |         |  $L = 3$   |             |        |  $L = 5$   |             |         |  $L = 10$  |             |        |  $L = 20$  |             |        |
> | ------- | :--------: | :---------: | :----: | :--------: | :---------: | :-----: | :--------: | :---------: | :----: | :--------: | :---------: | :----: |
> |         | IQL$_{.5}$ | IQL$_{.85}$ |  Time  | IQL$_{.5}$ | IQL$_{.85}$ |  Time   | IQL$_{.5}$ | IQL$_{.85}$ |  Time  | IQL$_{.5}$ | IQL$_{.85}$ |  Time  |
> | PIF$_s$ |   45.17    |    35.01    | 11.05  |   32.59    |    20.21    |  10.80  |   25.60    |    14.75    | 10.87  |   27.10    |    15.81    | 10.90  |
> |         |   (4.26)   |   (9.64)    | (0.33) |   (7.24)   |   (6.94)    | (0.29)  |   (6.17)   |   (3.80)    | (0.27) |   (6.12)   |   (3.06)    | (0.32) |
> | PIF$_e$ |   39.69    |    26.70    | 10.24  |   32.24    |    21.05    |  10.17  |   31.46    |    21.89    | 10.53  |   31.23    |    21.15    | 10.48  |
> |         |   (4.55)   |   (6.15)    | (0.27) |   (7.14)   |   (4.79)    | (0.39)  |   (6.17)   |   (3.92)    | (0.44) |   (5.83)   |   (4.20)    | (0.32) |
> | PIF$_q$ |   39.82    |    25.42    | 10.47  |   30.81    |    20.03    |  10.42  |   29.89    |    18.73    | 10.41  |   31.36    |    19.96    | 10.46  |
> |         |   (4.89)   |   (5.88)    | (0.21) |   (5.89)   |   (4.12)    | (0.29)  |   (4.47)   |   (3.53)    | (0.27) |   (5.91)   |   (3.56)    | (0.32) |
> | STVB    |   42.41    |    23.01    | 40.02  |   30.89    |    16.75    |  48.83  |   28.87    |    17.37    | 38.98  |   28.59    |    16.75    | 38.35  |
> |         |   (5.71)   |   (5.95)    | (1.20) |   (7.04)   |   (3.17)    | (37.64) |   (6.76)   |   (3.31)    | (0.65) |   (6.46)   |   (3.46)    | (1.04) |
> | VBPP    |   38.73    |    22.79    | 26.60  |   30.97    |    19.41    |  25.88  |   30.50    |    19.31    | 26.86  |   37.09    |    25.06    | 28.14  |
> |         |   (4.35)   |   (4.14)    | (0.74) |   (4.88)   |   (3.23)    | (0.70)  |   (5.37)   |   (3.67)    | (0.84) |   (6.90)   |   (3.57)    | (0.90) |
>
> **Table C:** Results on three types of synthetic data with standard errors in brackets.

---

> > ### Comment · Reviewer_jgS8 · 2021-09-02
> > **Response to authors**
> >
> > I would like to thank the authors for taking the time and effort to address my concerns through additional experimental results. Based on this, I am happy that my original score of 7 is an accurate reflection of this paper.
> >
> > As noted by reviewer SxYf, I agree that it is a real shame that code cannot be shared. Whilst the authors have made efforts to describe their algorithm, sharing code is still the gold standard for reproducibility. This has not impacted my score as I appreciate this is out of the authors' control, however, I feel it is still worth acknowledging.

---

### Official Review · Reviewer_rjT5 · 2021-07-19

**Rating:** 9
**Confidence:** 4

**Summary:**

This outstanding paper presents an extremely novel approach (for the machine learning community) to the Gaussian process formulation. This formulation is interesting in its own right. But, the paper goes beyond the pure GP interpretation aspect and introduces this formulation for a very specific and precise purpose: to handle the challenging Poisson process likelihood function. This likelihood function is non-i.i.d. as it involves and integral term (equation 2 of the submission). The integral formulation leads to sneaky analytical tricks on the Poisson likelihood, and provides a new way to handle the interesting "doubly stochastic" inference problem.

The authors import some highly technical and novel concepts from physics, to attack the latent GP intensity function of the Poisson process. This leads to a highly efficient and principled algorithm. Moreover, the high degree of novelty is such that the technique may be applied to other problems with latent GP intensity. For example, their method would also allow to handle the likelihood function of survival analysis with latent GP + link function for the hazard function, which also includes an integral term.

The writing is crisp and compelling. This was a pleasure to behold. The experiments are more than sufficient. It is impressive that the authors made this work in practice, and I hope they see fit to release their code, but if they don't it is okay and the paper more than stands on its own.


**Limitations And Societal Impact:**

Yes.

**Main Review:**

My very positive appraisal is in the overview section. Here I give some very minor constructive feedback.

My main comment regards the discussion of computational complexity. The abstract is slightly misleading, since the stated big O pertains to one iteration, but the number of iterations required clearly depends on N. It would be great to see a table of computational complexities per iteration, for the various alternatives algorithms. This could be broken down by "predictive mean, predictive variance, marginal likelihood", and in some cases with an alternative for primal and dual formulations. Also, the O(N^2) on line 256 is a little unclear as the methods under question can be implemented in primal or dual form (scaling in # basis functions of # data), and in the primal form do obtain a similar scaling to the proposed method.

These are minor constructive criticisms and would be trivial to handle in the camera ready version.

**Time Spent Reviewing:**

2

---

> ### Author Response · Authors · 2021-08-10
> **To Reviewer rjT5**
>
> We would like to thank the reviewer for the highly positive comments, by which we are strongly encouraged. Our work is based on interdisciplinary techniques, and we believe that it can spawn further development of ML community by introducing a methodology that has been overlooked so far. Below we provide a detailed response to each of the comments.
>
>
>
> **Table of computational complexities:**
>
> According to the reviewer's suggestion, we will add to the supplementary material a table of computational complexities per iteration for alternative algorithms (see Table A). Here we focus on the algorithms that scales linearly with the number of data points. Note that the complexity of each algorithm could be reduced by utilizing a stochastic gradient descent algorithm or exploiting the Kronecker structure.
>
>
>
> **About the complexity of [45] (line 256):**
>
> We thank the reviewer for pointing out our misunderstanding. Although the reference [45] seems not exploit the computation in primal form, it is possible for the method to be implemented in primal form, where the MAP estimation scales linearly with the data size as shown in Table A. We will correct the sentence on lines 255-256 appropriately. It should be noted that the term $N_{\text{mc}}L$ in our PIF's $O_v$ and $O_l$ in Table A disappears under the quadratic link function because the integral in Eq. (19) can be performed analytically.
>
>
> &nbsp;
>
> |                 |                          PIF                         |                  [45]                 | VBPP[31] & STVB[3] |     [24]    |
> |:---------------:|:----------------------------------------------------:|:-------------------------------------:|:------------------:|:-----------:|
> | $O_m$ |                      $(NL+L^2)Q$                     |       $(N\cdot\text{min}(N,L))Q$      |    $(NL^2+L^3)Q$   | $(NL+L^2)P$ |
> | $O_v$ | $O_m + (N+L)\cdot\text{min}(N^2,L^2)+N_{\text{mc}}L$ | $O_m + (N+L)\cdot\text{min}(N^2,L^2)$ |    $(NL^2+L^3)Q$   | $(NL+L^2)P$ |
> | $O_l$ | $O_m + (N+L)\cdot\text{min}(N^2,L^2)+N_{\text{mc}}L$ | $O_m + (N+L)\cdot\text{min}(N^2,L^2)$ |     $NL^2+L^3$     |   $NL+L^2$  |
>
>
> **Table A:** $O_m$, $O_v$, $O_l$ represent the computational costs of the MAP/predictive mean, the predictive covariance, and the marginal likelihood/evidence lower bound, respectively. $Q$ and $P$ represent the number of iterations and the number of MCMC samplings, respectively.

---

> > ### Comment · Reviewer_rjT5 · 2021-08-25
> > **thanks for ...**
> >
> > ... the information, especially the table !

---

### Decision · Program_Chairs · 2021-09-27

**Decision:**

Accept (Spotlight)

**Comment:**

This paper presents a path integral formulation for inference in Gaussian Cox process models, which is general for any prior kernel and a variety of link functions. Three out of four reviewers recommend acceptance, with one of the reviewers assessing the paper as a top 15% of accepted papers at NeurIPS. The main criticism of the paper (raised by Reviewer SxYf and shared by other reviewers and myself)  is that the lack of commitment from the authors to make the code publicly available will impede reproducibility. I agree with this point but I understand there are cases where this is not possible. I believe the technical contribution of the paper is strong enough to be beneficial to the NeurIPS community but encourage the authors to improve the reproducibility of their approach. For example, they can include an algorithm (in pseudocode) in the main paper detailing the exact computations required in their method. Besides addressing the reviewers’ concerns, I also encourage the authors to discuss the limitations of their approach a bit further, for example by comparing hyper-parameter estimation in their approach versus other methods such as those based on variational inference.